J Physiol 603.7 (2025) pp 2119–2138

# Biophysical modelling of intrinsic cardiac nervous system neuronal electrophysiology based on single-cell transcriptomics

Suranjana Gupta[1] , Michelle M. Gee[2,3] , Adam J. H. Newton[1], Lakshmi Kuttippurathu[2], Alison Moss[2], John D. Tompkins[4] , James S. Schwaber[2,3], Rajanikanth Vadigepalli[2,3] and William W. Lytton[1,5]

[1] Department of Physiology and Pharmacology, SUNY Downstate Health Sciences University, Brooklyn, NY, USA
[2] Daniel Baugh Institute for Functional Genomics/Computational Biology, Department of Pathology and Genomic Medicine, Thomas Jefferson University, Philadelphia, PA, USA
[3] Department of Chemical and Biomolecular Engineering, University of Delaware, Newark, DE, USA
[4] UCLA Cardiac Arrhythmia Center and Neurocardiology Research Program of Excellence, David Geffen School of Medicine at UCLA, University of California, Los Angeles, CA, USA
[5] Department of Neurology, Kings County Hospital, Brooklyn, NY, USA

Handling Editors: Natalia Trayanova & Eleonora Grandi

The peer review history is available in the Supporting Information section of this article (https://doi.org/10.1113/JP287595#support-information-section).

**Abstract figure legend** The intrinsic cardiac nervous system regulates the beat-to-beat function of the heart. Hodgkin–Huxley ion channel models from literature were selected based on the ion channels found in single-neuron transcriptomic data. The transcriptomic data were binarized to confer combinations of ion channel presence or absence for each neuron in a library of parallel conductance models. The model-predicted electrophysiological behaviour reflects the distribution of firing patterns observed experimentally. These models are a first step towards bridging the gap between single-cell transcriptomic data and predictive models of physiology.

S. Gupta and M. M. Gee are co-first authors.

The Journal of Physiology

**Abstract**  The intrinsic cardiac nervous system (ICNS), termed as the heart's 'little brain', is the final point of neural regulation of cardiac function. Studying the dynamic behaviour of these ICNS neurons via multiscale neuronal computer models has been limited by the sparsity of electrophysiological data. We developed and analysed a computational library of neuronal electrophysiological models based on single neuron transcriptomic data obtained from ICNS neurons. Each neuronal genotype was characterized by a unique combination of ion channels identified from the transcriptomic data, using a cycle threshold cutoff that ensured the electrical excitability of the neuronal models. The parameters of the ion channel models were grounded based on passive properties (resting membrane potential, input impedance and rheobase) to avoid biasing the dynamic behaviour of the model. Consistent with experimental observations, the emergent model dynamics showed phasic activity in response to the current clamp stimulus in a majority of neuronal genotypes (61%). Additionally, 24% of the ICNS neurons showed a tonic response, 11% were phasic-to-tonic with increasing current stimulation and 3% showed tonic-to-phasic behaviour. The computational approach and the library of models bridge the gap between widely available molecular-level gene expression and sparse cellular-level electrophysiology for studying the functional role of the ICNS in cardiac regulation and pathology.

(Received 30 August 2024; accepted after revision 14 February 2025; first published online 11 March 2025)
**Corresponding author** R. Vadigepalli: Daniel Baugh Institute for Functional Genomics/Computational Biology, Department of Pathology and Genomic Medicine, Thomas Jefferson University, Philadelphia, PA, USA.    Email: Rajanikanth.Vadigepalli@jefferson.edu

**Key points**

- Computational models were developed of neuron electrophysiology from single-cell transcriptomic data from neurons in the heart's 'little brain': the intrinsic cardiac nervous system.
- The single-cell transcriptomic data were thresholded to select the ion channel combinations in each neuronal model.
- The library of neuronal models was constrained by the passive electrical properties of the neurons and predicted a distribution of phasic and tonic responses that aligns with experimental observations.
- The ratios of model-predicted conductance values are correlated with the gene expression ratios from transcriptomic data.
- These neuron models are a first step towards connecting single-cell transcriptomic data to dynamic, predictive physiology-based models.

## Introduction

Parasympathetic and sympathetic imbalance contributes to the aetiology of many cardiovascular diseases. A key regulator of sympathovagal balance is the heart's 'little brain', the intrinsic cardiac nervous system (ICNS), which contains both cholinergic and catecholaminergic neurons (Armour, 2008; Hadaya & Ardell, 2020; Hanna et al., 2021; Moss et al., 2021). As the final neural regulatory point for the heart, the ICNS mediates the

**Rajanikanth Vadigepalli** is currently a Professor and Vice Chair of Research in Pathology and Genomic Medicine at Thomas Jefferson University, Philadelphia, PA, USA. He received his Bachelor's in Chemical Engineering from the Indian Institute of Technology–Madras and a PhD in Chemical Engineering from the University of Delaware in 2001, with a Specialization in Systems and Control Engineering. His collaborative research programme is driven by a convergence of systems engineering, computational modelling, bioinformatics, artificial intelligence and single-cell scale molecular omics to identify and target key control points for intervention in chronic disease. Ongoing projects in his team are focused on central and peripheral neural circuits controlling the heart and brainstem neuroimmune processes leading to hypertension, liver regeneration in alcohol-associated liver disease and cell fate regulation underlying developmental defects.

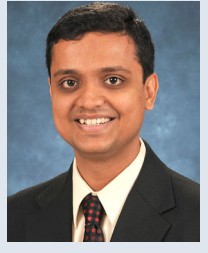

balance of parasympathetic and sympathetic inputs to the cardiac tissue. Neural remodelling within the ICNS has been linked to the progression of cardiovascular disease (Beaumont et al., 2016; Salavatian et al., 2016; Vaseghi et al., 2017). Both phasic and tonic firing responses have been observed in mice, pigs and humans, and neural remodelling to modulate these behaviours could regulate sympathovagal balance in health and disease (Tompkins et al., 2025).

Phasic and tonic electrophysiological behaviour arises from a multitude of ion channel combinations through a complex mapping relating variable molecular expression to relatively more constrained functional responses. Recently, the increased availability of high-throughput, single-neuron transcriptomics has made it possible to identify the exact combinations of ion channels present in each cell to connect subcellular components to cellular function (Moss et al., 2021). These transcriptomic datasets have been mined to address questions of how ion channel degeneracy contributes to firing robustness in neurons (Drion et al., 2015; Foster et al., 1993; Goaillard & Marder, 2021; Nandi et al., 2022; Roy & Narayanan, 2023), but have not been used to study how high-dimensional ion channel expression collapses to a restricted set of phasic and tonic responsive phenotypes. Differences in ion channel conductances that drive transitions from phasic to tonic firing in a neuron are potential regulatory points for controlling sympathovagal balance in the ICNS.

To address this question, we use the well-established Hodgkin–Huxley models and combine them with single-cell transcriptomic data to identify specific ion channel combinations for each neuron. This computational approach allows us to explore the contributions of individual ion channels that would not be possible without inferring channel involvement through time-consuming pharmacological blockades or without assuming channel types (Schwaber et al., 1993; Shevtsova et al., 2020). Instead, *in silico* screening can be performed to identify the most important ion channels for further experimental testing. In addition, neurons of the same cell type have electrophysiological behaviour consistent with each other in response to current clamp stimulus, but vary in their ion channel conductance densities (Goaillard & Marder, 2021). This heterogeneity may contribute to the variable responses of neurons of the same type to perturbations, muddling the association between an ion channel and a particular function identified via conventional experimental approaches (Goaillard & Marder, 2021). Electrophysiological recording of neuronal electrical activity has been a productive approach to studying the ICNS to capture neuronal firing rate and membrane electrical behaviour, but it is labour-intensive and, therefore, low throughput. More recently, systems biology provides a complementary approach by capitalizing on high-throughput trans-

criptomic techniques (Hanna et al., 2021; Moss et al., 2021) that are becoming increasingly available through data-sharing initiatives, such as the National Institutes of Health's SPARC programme (https://sparc.science/).

In this work, we aim to connect the electrophysiological behaviour of ICNS neurons to their gene expression using transcriptomics-based single-cell parallel conductance Hodgkin–Huxley neuronal genotype computational models. We present a strategy for using single-neuron transcriptomic data to predict neuronal membrane physiology, demonstrating a workflow for building a library of neuronal genotype models. We used data from 321 porcine right atrial ganglionic plexus (RAGP) neurons to deduce the presence or absence of particular channel types in each neuron. We then used Hodgkin–Huxley ion-channel models from open-source model repositories to construct a library of parallel conductance models reflecting ion channel combinations and predicting electrophysiological behaviour.

## Methods

We propose a six-step workflow for the development of electrophysiological neuronal models from single-neuron transcriptomic data (Fig. 1). We expand upon steps I and V in 'Morphology, physiology and transcriptomics of neurons', step III in 'Ion channel model selection' and step IV in 'Parameter estimation'.

### Morphology, physiology and transcriptomics of neurons

The morphology of Yucatán minipig RAGP principal neurons (PN) was obtained from previously reported experimental data (Hanna et al., 2021). Porcine RAGP PN somata are generally elliptical with a radius spanning 15–30 μm along their minor axis and 20–47 μm along their major axis (Hanna et al., 2021; Moss et al., 2021). The typical minipig RAGP neuron cross-sectional area is ∼1400 μm$^2$ and ranges from 600 to 4000 μm$^2$ (Hanna et al., 2021). In our single-neuron models, we used a 21 μm diameter and 21 μm length to achieve this area using the NEURON software's cylindrical section (Awile et al., 2022; Carnevale & Hines, 2006).

Neuronal models of RAGP PN were constrained on the basis of passive electrical properties reported for Yucatán minipig (Hanna et al., 2021), guinea pig (Edwards et al., 1995), rat (Selyanko, 1992) and mouse ICNS (Harper & Adams, 2021; Lizot et al., 2022). These properties were: (1) resting membrane potential (RMP) near −60 mV (Hanna et al., 2021); (2) input impedance ($R_{in}$) 40–300 MΩ (Edwards et al., 1995; Hanna et al., 2021; Harper & Adams, 2021; Selyanko, 1992); (3) rheobase 0.02–0.08 nA (Lizot et al., 2022); and (4) leak reversal potential ($E_{pas}$) of −80 to −50 mV, corresponding to a range between $E_K$ and $E_h$.

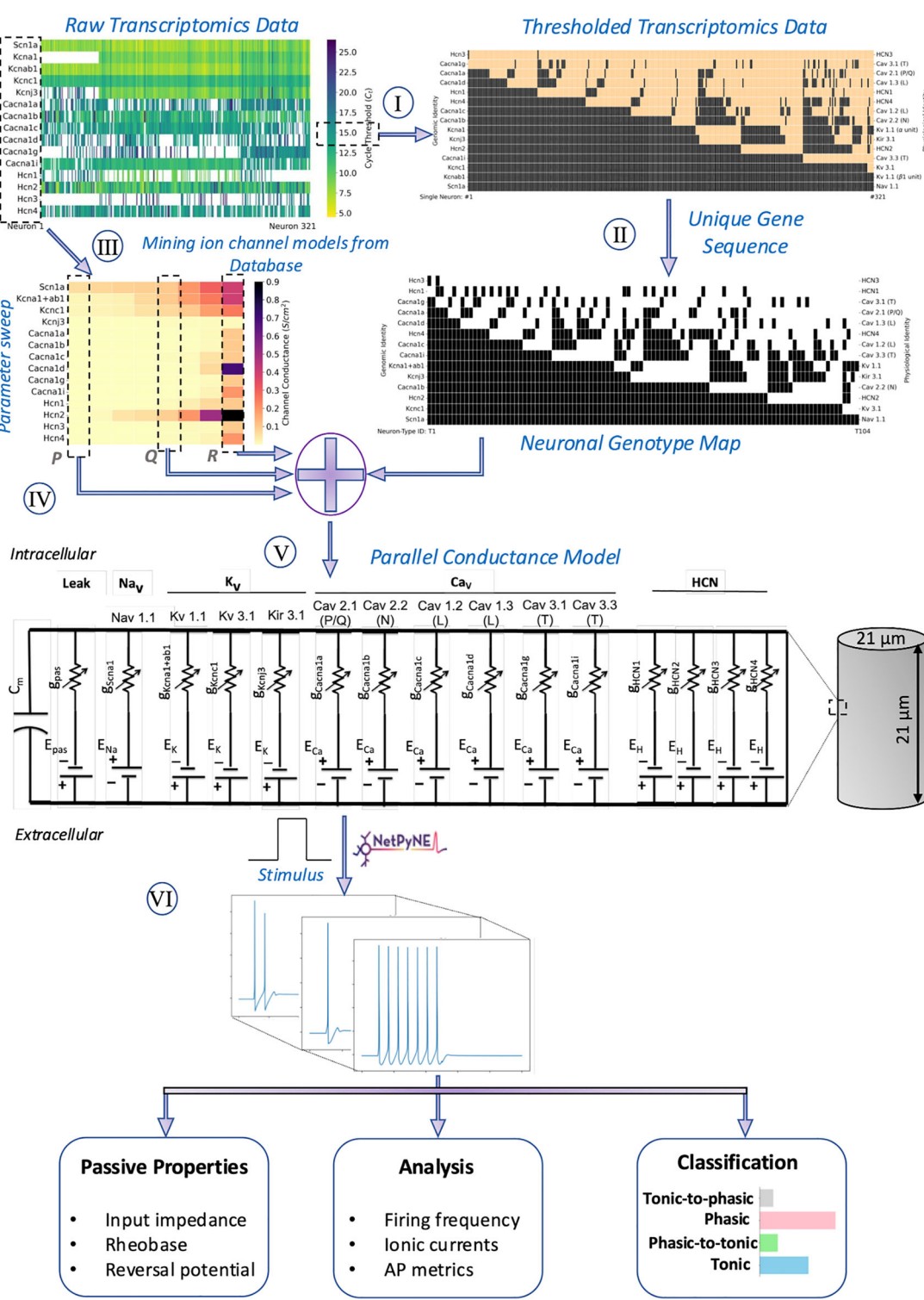

**Figure 1. Workflow for development of electrophysiological models starting with single neuron gene expression data and model database of ion channel kinetics**

(I) Single-neuron transcriptomic data were thresholded using a cycle threshold cutoff to select ion channel presence or absence for neuron models. (II) Unique neuronal genotypes were identified by removing redundant ion channel combinations. (III) Ion channel models corresponding to genes identified in the transcriptomic data were curated from public databases. (IV) Fixed conductance values were selected for each ion channel. (V) Morphological properties of ICNS neurons were incorporated to construct a library of parallel conductance models. (VI) Model responses to the current clamp stimulus were analysed and classified. [Colour figure can be viewed at wileyonlinelibrary.com]

Previously published high throughput quantitative polymerase chain reaction (HT-qPCR) data from 405 single RAGP neurons and 15 mRNA transcripts coding for 14 ion channel genes were used to select ion channel presence or absence in single-neuron models (Moss et al., 2021; see Data availability statement). Each mRNA transcript codes for one ion channel or ion channel subunit. The co-expression of the transcripts *Kcna1* and its subunit *ab1* were translated into equivalent biophysics, as expanded in the subsequent section. Samples were collected from two male and two female Yucatán minipigs using laser capture microdissection (Moss et al., 2021). After examination of the dataset for quality control based on the presence of $Na^+$ channel expression, samples from one female pig were removed, leaving samples from 321 neurons.

Owing to a gradient in ion channel gene expression, we binarized the data to identify ion channels to include in each neuronal model (Fig. 2). A cycle threshold ($C_t$), which is inversely correlated to gene expression level, was selected as the threshold metric and a value of 15 cycles was found to be a suitable cut-off. Analysis of the effect of $C_t$ threshold on the distribution of neuronal genotypes and on the electrophysiological behaviour in our population of neuronal genotype models was assessed to justify the selection of 15 as a $C_t$ threshold (Fig. 3). After thresholding with our selected $C_t$ threshold of 15, we identified 104 unique combinations of ion channels from the 321 single-neurons. We refer to each unique combination of ion channels as a *neuronal genotype*.

## Ion channel model selection

Ion channel models for each of the 14 ion channels were selected from three public databases: Channelpedia (channelpedia.epfl.ch; Ranjan et al., 2011), ModelDB modeldb.science; McDougal et al., 2017), and Ion Channel Genealogy (icg.neurotheory.ox.ac.uk; Podlaski et al., 2017). An in-house library of ion channel models mined from the public databases and literature survey was used to track ion channel model properties, gating kinetics, physiological function, experimental protocol and tissues/cells for model creation. We employed the database to establish provenance and to compare each ion channel isoform model against its counterparts. The initial selection of ion channel models from the databases relied on identifying the extent to which the model could be assigned to a particular gene rather than being a generic model. Multiple ion channel models of the same genotype, mined from the different databases, were compared on the basis of their activation and inactivation dynamics to assess their regions of operation. We used Hodgkin–Huxley parallel conductance models (Table 1) with conductance values for $Na^+$, $K^+$ and HCN ion channels (Table 2).

We simulated channel combinations to find an ion channel model for each gene that worked in the full parallel conductance model. The choice of a suitable sodium (*Scn1a*, Nav1.1) channel is particularly important due to its role in spiking. Multiple kinetic models were available, even within a single database. We identified five potential Nav1.1 models (Channelpedia ID no. 35, ModelDB Accession no. 20756, no. 256632, no. 264834 and $Na^+$ model reported in Rybak et al., 1997). We ran comparative simulations to find one model for each ion channel that yielded physiologically stable responses across a range of conductance values. The criteria for selection included $E_{pas}$, $R_{in}$ and rheobase within the range of experimentally measured data (Edwards et al., 1995; Hanna et al., 2021; Harper & Adams, 2021; Selyanko,

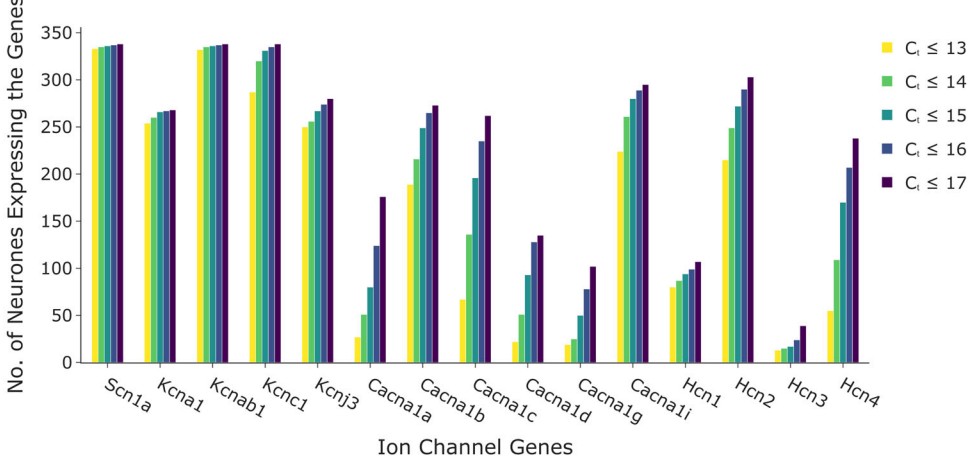

**Figure 2. Selection of expression threshold for filtering transcriptomic data**
Number of neurons identified with each ion channel transcript at $C_t$ values from 13 to 17. $C_t \leq 15$ was chosen to denote ion channel presence. [Colour figure can be viewed at wileyonlinelibrary.com]

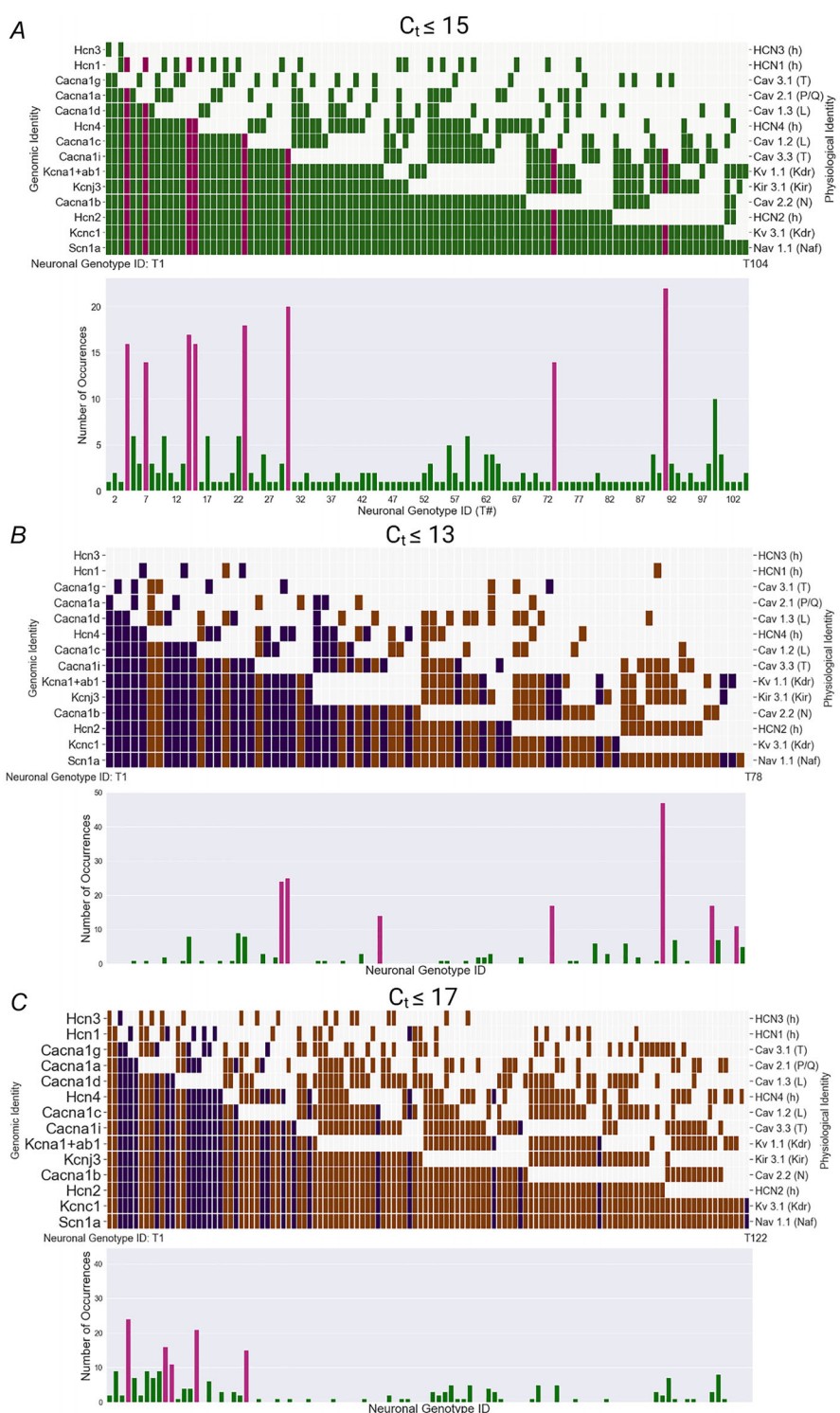

**Figure 3. Neuronal genotypes resulting from thresholded single neuron gene expression data on ion channels**

*A*, top, binary map for the 104 unique channel combinations ordered by frequency of occurrence of ion channels when using a cycle threshold $C_t \leq 15$. Bottom, number of cells of each neuronal genotype (321 cells, 104 neuronal genotypes). The eight common neuronal genotypes are highlighted in magenta, while the remaining neuronal genotypes are in green. Neuronal genotypes were considered to be common if there were more than 10 occurrences. *B* and *C*, neuronal genotypes and frequency of occurrence for $C_t \leq 13$ (*B*) and $C_t \leq 17$ (*C*). New neuronal genotypes not defined by thresholding transcriptomic data with $C_t \leq 15$ are highlighted in brown. [Colour figure can be viewed at wileyonlinelibrary.com]

**Table 1. Kinetic model of ion channels employed in single-compartment RAGP principal neuron models.**

| Ion channel | Model equations | Governing equations of parameter |
|---|---|---|
| Nav 1.1 (*Scn1a*) | $i_{Scn1a} = g_{Scn1a} \times (v_m - E_{Na})$ <br> $g_{Scn1a} = \bar{g}_{Scn1a} \times m^3 \times h$ | $\alpha_m = \dfrac{0.182 \times (v_m+35)}{1-e^{-\frac{(v_m+35)}{9}}}$ <br> $\beta_m = \dfrac{0.124 \times (-v_m-35)}{1-e^{-\frac{(v_m+35)}{9}}}$ <br> $m_\infty = \dfrac{\alpha_m}{\alpha_m+\beta_m}$ <br> $\tau_m = \dfrac{1}{\alpha_m+\beta_m}$ <br> $h_\infty = \dfrac{1}{1+e^{\frac{(v_m+65)}{6.2}}}$ <br> $\tau_h = \dfrac{1}{\frac{0.024 \times (v_m+50)}{1-e^{\frac{-(v_m+50)}{5}}} + \frac{0.0091 \times (-v_m-75.000123)}{1-e^{\frac{-(-v_m-75.000123)}{5}}}}$ |
| HCN1 | $i_{Hcn1} = \bar{g}_{Hcn1} \times m \times (v_m - E_{Hcn})$ | $m_\infty = \dfrac{1}{1+e^{\frac{(v_m+94)}{8.1}}}$ <br> $\tau_m = 30$ ms |
| HCN2 | $i_{Hcn2} = \bar{g}_{Hcn2} \times m \times (v_m - E_{Hcn})$ | $m_\infty = \dfrac{1}{1+e^{\frac{(v_m+99)}{6.2}}}$ <br> $\tau_m = 184$ ms |
| HCN3 | $i_{Hcn3} = \bar{g}_{Hcn3} \times m \times (v_m - E_{Hcn})$ | $m_\infty = \dfrac{1}{1+e^{\frac{(v_m+96)}{8.6}}}$ <br> $\tau_m = 265$ ms |
| HCN4 | $i_{Hcn4} = \bar{g}_{Hcn4} \times m \times (v_m - E_{Hcn})$ | $m_\infty = \dfrac{1}{1+e^{\frac{(v_m+100)}{9.6}}}$ <br> $\tau_m = 461$ ms |
| Kv 1.1 (*Kcna1* + *Kcnab1*) | $i_{Kcna1+ab1} = g_{Kcna1+ab1} \times (v_m - E_K)$ <br> $g_{Kcna1+ab1} = \bar{g}_{Kcna1+ab1} \times n^4 \times x$ | $\alpha_n = 0.12889 \times e^{\frac{-(v_m+45)}{-33.90877}}$ <br> $\beta_n = 0.12889 \times e^{\frac{-(v_m+45)}{12.42101}}$ <br> $n_\infty = \dfrac{\alpha_n}{\alpha_n+\beta_n}$ <br> $\tau_n = \dfrac{1}{4.171167511 \times (\alpha_n+\beta_n)}$ <br> $x_\infty = \dfrac{0.95}{(1+e^{\frac{(v_m+59)}{3}})^{0.5}} + 0.05$ <br> $\tau_x = \dfrac{500}{14 \times e^{\frac{(v_m+28)}{20}} + 29 \times e^{\frac{-(v_m+28)}{10}}} + 6$ |
| Kv 3.1 (*Kcnc1*) | $i_{Kcnc1} = g_{Kcnc1} \times (v_m - E_K)$ <br> $g_{Kcnc1} = \bar{g}_{Kcnc1} \times (\phi \times n^2) + ((1-\phi) \times p)$ | $n_\infty = \dfrac{1}{\sqrt{1+e^{\frac{-(v_m+15)}{5}}}}$ <br> $p_\infty = \dfrac{1}{1+e^{\frac{-(v_m+23)}{6}}}$ <br> $\tau_n = \dfrac{100}{11 \times e^{\frac{(v_m+60)}{24}} + 21 \times e^{\frac{-(v_m+60)}{23}}} + 0.7$ <br> $\tau_p = \dfrac{100}{4 \times e^{\frac{(v_m+60)}{32}} + 5 \times e^{\frac{-(v_m+60)}{22}}} + 5$ |
| Kir 3.1 (*Kcnj3*, GIRK-1) | $i_{Kcnj3} = g_{Kcnj3} \times (v_m - E_K)$ <br> $g_{Kcnj3} = \bar{g}_{Kcnj3} \times 2.716898432 \times n$ | $\alpha_n = 0.001 \times \dfrac{(v_m+30)}{1-e^{\frac{(v_m+30)}{9}}}$ <br> $\beta_n = -0.001 \times \dfrac{(v_m+30)}{1-e^{\frac{(v_m+30)}{9}}}$ <br> $n_\infty = \dfrac{\alpha_n}{\alpha_n+\beta_n}$ <br> $\tau_n = \dfrac{1/2.716898432}{\alpha_n+\beta_n}$ |
| P/Q-type (*Cacna1a*, Cav 2.1) | $i_{Cacna1a} = \bar{g}_{Cacna1a} \times m \times \Phi_{Ca}$ | $\alpha_m = \dfrac{8.5}{1+e^{\frac{(v_m-8)}{-12.5}}}$ <br> $\beta_m = \dfrac{35}{1+e^{\frac{(v_m+74)}{14.5}}}$ <br> $m_\infty = \dfrac{\alpha_m}{\alpha_m+\beta_m}$ <br> $\tau_m = \dfrac{1}{\alpha_m+\beta_m}$ |
| N-type (*Cacna1b*, Cav 2.2) | $i_{Cacna1b} = g_{Cacna1b} \times \Phi_{Ca}$ <br> $g_{Cacna1b} = \bar{g}_{Cacna1b} \times m^2 \times h$ | $\alpha_m = \dfrac{0.1 \times (v_m-20)}{1-e^{\frac{-(v_m-20)}{10}}}$ <br> $\beta_m = 0.4 \times e^{\frac{-(v_m+25)}{18}}$ <br> $\alpha_h = 0.01 \times e^{\frac{-(v_m+50)}{10}}$ <br> $\beta_h = \dfrac{0.1}{1+e^{\frac{-(v_m+17)}{17}}}$ <br> $i_\infty = \dfrac{\alpha_i}{\alpha_i+\beta_i} \quad i = m, h$ <br> $\tau_i = \dfrac{1}{\alpha_i+\beta_i} \quad i = m, h$ |

*(Continued)*

**Table 1. (Continued)**

| Ion channel | Model equations | Governing equations of parameter |
|---|---|---|
| L-type (*Cacna1c*, Cav 1.2) | $i_{Cacna1c} =$ $g_{Cacna1c} \times \Phi_{Ca}$ $g_{Cacna1c} =$ $\bar{g}_{Cacna1c} \times m^2 \times h$ | $m_\infty = \dfrac{1}{1+e^{\frac{(v_m+30.000)}{-6}}}$ $\tau_m = 10 \; ms$ $h_\infty = \dfrac{1}{1+e^{\frac{(v_m+80.000)}{6.4}}}$ $\tau_h = 59 \; ms$ |
| L-type (*Cacna1d*, Cav 1.3) | $i_{Cacna1d} =$ $g_{Cacna1d} \times \Phi_{Ca}$ $g_{Cacna1d} =$ $\bar{g}_{Cacna1d} \times m^2 \times h$ | $m_\infty = \dfrac{1}{1+e^{\frac{(v_m+33)}{-6.7}}}$ $h_\infty = \dfrac{1}{1+e^{\frac{(v_m+13.4)}{11.9}}}$ $\alpha_m = \dfrac{0.0398\times(v_m+8.124)}{e^{\frac{(v_m+8.124)}{9.005}}-1}$ $\beta_m = 0.99 \times e^{\frac{v_m}{31.4}}$ $\tau_m = \dfrac{1}{3\times(\alpha_m+\beta_m)}$ $\tau_h = 44.3/3 \; ms$ |
| T-type (*Cacna1g*, Cav 3.1) | $i_{Cacna1g} =$ $g_{Cacna1g} \times \Phi_{Ca}$ $g_{Cacna1g} =$ $\bar{g}_{Cacna1g} \times m \times h$ | $m_\infty = \dfrac{1}{1+e^{\frac{(v_m+42.921064)}{-5.163208}}}$ $\tau_m = \begin{cases} -0.855809 + (1.493527 \times e^{\frac{-v_m}{27.414182}}) & v_m < -10 \\ 1.0 \; ms & v_m \geq -10 \end{cases}$ $h_\infty = \dfrac{1}{1+e^{\frac{(v_m+72.907420)}{4.575763}}}$ $\tau_h = 9.987873 + (0.002883 \times e^{\frac{-v_m}{5.598574}})$ |
| T-type (*Cacna1i*, Cav 3.3) | $i_{Cacna1i} =$ $g_{Cacna1i} \times \Phi_{Ca}$ $g_{Cacna1i} =$ $\bar{g}_{Cacna1i} \times m^2 \times h$ | $m_\infty = \dfrac{1}{1+e^{\frac{-(v_m+59)}{6.2}}}$ $h_\infty = \dfrac{1}{1+e^{\frac{(v_m+83)}{4.0}}}$ $\tau_m = 0.612 + \dfrac{1}{\frac{e^{\frac{-(v_m+134)}{16.7}}+e^{\frac{(v_m+18.8)}{18.2}}}{6.898648307}}$ $\tau_h = \begin{cases} \dfrac{e^{\frac{(v_m+469)}{66.6}}}{3.737192819} & v_m < -82 \\ \dfrac{28+e^{\frac{-(v_m+24)}{10.5}}}{3.737192819} & v_m \geq -82 \end{cases}$ |

Goldman–Hodgkin–Katz (GHK) flux equation for Cav channels. For ion S, where $z_S$ is the charge, $[S]_i$ and $[S]_o$ in mM, $F$ Faraday's constant (C/mol), $R$ is the gas constant (J/(mol K)), $T$ is the temperature (K) and $P_S$ is permeability of the membrane (cm/s). The $Ca^{2+}$ concentrations in the model remain constant with $[Ca]_i = 50$ nM and $[Ca]_o = 2$ mM.

$$\Phi_S = u_S = P_S \, z_S 10^{-3} \frac{v_m F}{RT} F z_S 10^{-3} ([S]_i f(-u_S) - [S]_o f(u_S))$$

where

$$f(x) = \begin{cases} 1 - \frac{x}{2} & if \; |x| < 10^{-4} \\ \frac{x}{\exp(x)-1} & otherwise \end{cases}.$$

1992). Three of the five models identified were not suitable, usually due to a large window current, which required an unphysiologically large $E_{pas}$ to compensate for the current at rest, or a large $R_{in}$ that produced a very leaky cell (Fig. 4*A*). To choose between the remaining two models, we then assessed the rheobase. Various experimental studies have noted a rheobase near 100 pA (Edwards et al., 1995; Hanna et al., 2021; Harper & Adams, 2021; Selyanko, 1992). A comparison of the two $Na^+$ channel candidates in the full parallel conductance model identified Channelpedia ID no. 35 as a better fit (Fig. 4*A*).

Potassium channel selection was also important for ensuring the possibility of neuronal excitability. *Kcna1* (Kv 1.1) is a delayed rectifier potassium (KDR) channel, which is non- or slowly-inactivating (time scale of seconds) (Song, 2002). Our data set a showed robust expression of *Kcna1* ($\alpha$ subunit of Kv 1.1). There was also a dominant

expression of *Kcnab1* ($\beta1$ regulatory subunit) across all neuronal genotypes. The subunit *Kcnab1* has been reported to confer fast inactivation in these channels (Allen et al., 2020; Heinemann et al., 1996; Rettig et al., 1994; Sewing et al., 1996). To account for the electrophysiological effect of the $\beta1$ subunit, a fast inactivation variable was introduced to our selected *Kcna1* KDR model (ModelDB Accession no. 80769). The model for *Kcnc1* (Kv 3.1), the most expressed potassium channel gene, was adapted from Rothman & Manis (2003a,b,c). The ion channel was fit to a two-component model with fast and slow activation processes, the relative contribution of which was established by a fractional amplitude parameter ($\phi$).

Similar to potassium channels, each variant of calcium channels possesses unique activation and inactivation kinetics. To ensure that we employ ion channel models

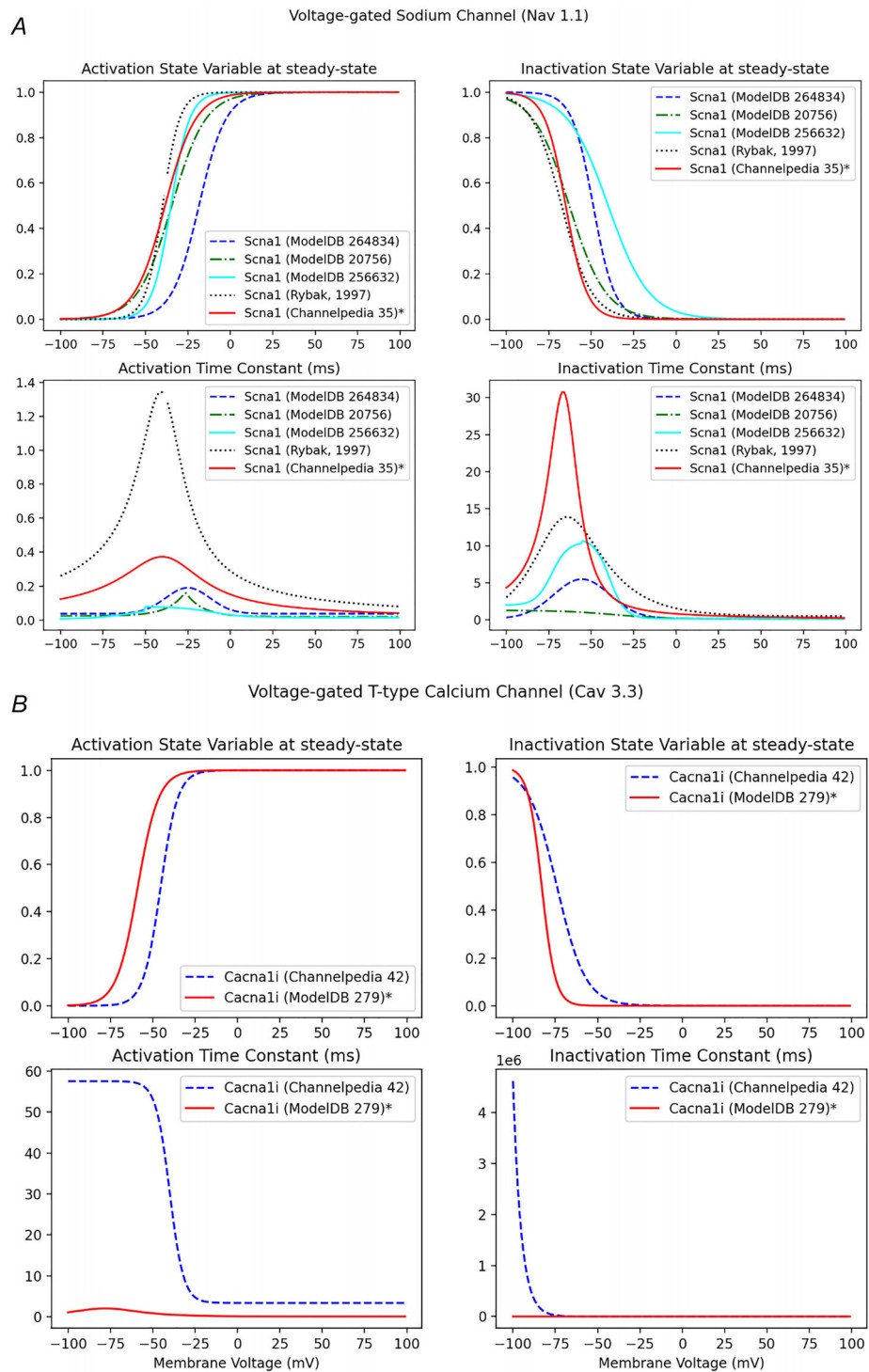

**Figure 4. Comparison of activation and inactivation curves for alternative ion channel models**

*A*, five alternative Nav1.1 models are compared, of which three were unsuitable due to a large window current which required an unphysiologically large reversal potential to compensate for the current at rest, or a leaky cell (large $R_{in}$). To choose between the remaining two models (Channelpedia ID no. 35 and ModelDB Accession no. 264834), we assessed the rheobase. A comparison of the two Na$^+$ channel candidates in the full parallel conductance model identified Channelpedia ID no. 35 as a better fit (marked by an *). *B*, the Cav3.3 models compared are Channelpedia ID no. 42 and ModelDB Accession no. 279. Channelpedia ID no. 42 generated a large window current around the RMP, making ModelDB Accession no. 279 more suitable for the current model (marked by an *). [Colour figure can be viewed at wileyonlinelibrary.com]

**Table 2. Estimated model parameters.**

| Parameter | Value | Literature model value |
|---|---|---|
| $\bar{g}_{Kcnj3}$ | $3.5 \times 10^{-3}$ S/cm$^2$ | $1 \times 10^{-3}$ S/cm$^2$ |
| $\bar{g}_{Kcna1+ab1}$ | 0.018 S/cm$^2$ | 0.011 S/cm$^2$ |
| $\bar{g}_{Kcnc1}$ | 0.018 S/cm$^2$ | 0.011 S/cm$^2$ |
| $\phi_{Kcnc1}$ | 0.2 | 0.85 |
| $\bar{g}_{Cacna1a}$ | 0.00005 S/cm$^2$ | $1 \times 10^{-5}$ S/cm$^2$ |
| $\bar{g}_{Cacna1b}$ | 0.0001 S/cm$^2$ | $1 \times 10^{-5}$ S/cm$^2$ |
| $\bar{g}_{Cacna1c}$ | 0.006 S/cm$^2$ | $1 \times 10^{-5}$ S/cm$^2$ |
| $\bar{g}_{Cacna1d}$ | 0.00045 S/cm$^2$ | $1.7 \times 10^{-6}$ S/cm$^2$ |
| $\bar{g}_{Cacna1g}$ | 0.0003 S/cm$^2$ | $1 \times 10^{-5}$ S/cm$^2$ |
| $\bar{g}_{Cacna1i}$ | 0.0006 S/cm$^2$ | $2 \times 10^{-4}$ S/cm$^2$ |
| $\bar{g}_{Hcn1}$ | 0.003 S/cm$^2$ | $1 \times 10^{-5}$ S/cm$^2$ |
| $\bar{g}_{Hcn2}$ | 0.009 S/cm$^2$ | $1 \times 10^{-5}$ S/cm$^2$ |
| $\bar{g}_{Hcn3}$ | 0.01 S/cm$^2$ | $1 \times 10^{-5}$ S/cm$^2$ |
| $\bar{g}_{Hcn4}$ | 0.0035 S/cm$^2$ | $1 \times 10^{-5}$ S/cm$^2$ |
| $\bar{g}_{Scn1a}$ | 0.075 S/cm$^2$ | $1 \times 10^{-5}$ S/cm$^2$ |

that most aptly describe their biophysics, we opted to select the models from Channelpedia. Four out of the six voltage-gated calcium channel models were taken from Channelpedia. Cav 1.3 was taken from ModelDB since its model in Channelpedia was unavailable. The Channelpedia model of Cav 3.3 generated a large window current around the RMP, which is uncharacteristic of a T-type voltage-gated calcium channel, which is active at voltages negative to the RMP (Fig. 4*B*). Owing to this inconsistency, the Cav 3.3 model was selected from ModelDB. The Cav 1.3 and 3.3 models were selected from ModelDB based on the cell type/species, experiments conducted and provenance, as mapped out by Ion Channel Genealogy.

Calcium channels used the Goldman–Hodgkin–Katz (GHK) flux equation with a Maclaurin series expansion of the voltage-dependent terms for numerical stability (Hille, 1991). Intracellular and extracellular potassium and sodium concentrations are tightly regulated during a spike such that they are maintained within the same order of magnitude, which results in their Nernst potentials remaining largely a constant. On the other hand, intracellular calcium concentration rises ~10-fold during a spike. Extracellular calcium concentration is in the millimolar order, while basal intracellular calcium concentration is of the order of hundreds of nanomolar. The calcium transient, which underlies an action potential, causes intracellular calcium concentration to rise to the order of micromolar, which results in a ~30 mV change in its Nernst potential. To ensure that this change in electrochemical driving force is accounted for in our models, we explicitly incorporated the GHK model for calcium channels.

## Parameter estimation

Once the ion channel models were chosen, simulations were performed for each ion channel to narrow the range of conductances so that the ensemble model's behaviour was physiologically stable. Ion channel conductances were initialized to their default values that came from either an original voltage clamp study in foreign tissue or a published ion channel model. A range of conductances around the default values for each ion channel model was set. Maximal conductance values were constrained sequentially in the order of Na$^+$, K$^+$ (*Kcna1*, *Kcnc1*, *Kcnj3*), Ca$^{2+}$ (a, b, c, d, g, i), and HCN (1, 2, 3, 4) channels. This order was selected based on which ion channels are known to contribute most significantly to electrophysiological behaviour.

A randomly sampled conductance matrix was chosen from the conductance ranges of respective ion channel models. Preliminary simulations were run to evaluate their passive properties. Combinations of conductances that did not yield physiologically tenable RMPs, $R_{in}$ and rheobase were rejected. After the stable range for each conductance value was identified, we randomly sampled within this range to produce six models. From six models, we identified three parameter sets with input impedance ($R_{in}$), reversal potential ($E_{pas}$) and rheobase within experimentally observed ranges (Edwards et al., 1995; Hanna et al., 2021; McAllen et al., 2011).

Channel conductances were primarily constrained by the passive electrical properties of the neurons. We had an additional degree of freedom in the fractional amplitude parameter ($\phi_{Kcnc1}$), which represents the relative contributions of the fast and slow activation processes in the Kcnc1 channel. We observed this parameter to mainly control the firing rates (code and example analysis available on GitHub, https://github.com/Daniel-Baugh-Institute/BiophysicalModellingOfIntrinsicCardiacNeurons). We varied $\phi_{Kcnc1}$ over a range of stimulus currents, and settled on a value that restricted the firing rates in accordance with experimental data (Harper & Adams, 2021; Vaseghi et al., 2017).

Simulations were run using the three-parameter sets identified. Additional tuning of the channel conductances was required in cases where ensemble models were dominated by artificial firing patterns. These patterns included continued firing without the application of a stimulus, sustained firing activity post-removal of the stimulus and incomplete repolarization resulting in elevated RMP for prolonged periods of time.

Classically, physiological arrest mechanisms exist to establish the reliable operation of the cells' electrical machinery. This ensures the operating points are always stable. This naturally embedded property needed to be explicitly modelled in our framework and was

**Table 3. Model parameters from literature.**

| Parameter | Value | Parameter | Value |
|---|---|---|---|
| Simulation time | 1000 ms | Time step (d$t$) | 25 μs |
| $V_{rmp}$ | −61 mV | Membrane threshold | −10 mV |
| Soma length | 21 μm | Soma diameter | 21 μm |
| $C_m$ | 1 μF/cm$^2$ | $R_a$ | 35.4 Ω-cm |
| No. of segments ($n_{seg}$) | 1 | Temperature | 35°C |
| $E_{Na}$ | 50 mV | $E_K$ | −77 mV |
| $E_h$ | −45 mV | | |

done via additional tuning of channel conductances. In this paper, we report a single parameter set that was most similar to the experimentally observed electrophysiological behaviour (Tables 2 and 3).

## Modelling and simulation tools

The model was implemented using NEURON v8.0 (http://neuron.yale.edu/) and the NetPyNE v1.0.0.2 Sobol branch (http://netpyne.org/) (Dura-Bernal et al., 2019). These modelling tools facilitated parallel simulations on high-performance computing platforms, allowing us to run over 400,000 simulations during model development. Please see the Data availability statement for all model and analysis files.

## Results

The model development workflow builds on ion channel expression data along with public resources on ion channel kinetics towards developing integrated electrophysiology models of ICNS neurons (Fig. 1). HT-qPCR data from single neurons on ion channel genes were used to select ion channel presence or absence in single-neuron models (step I) (Moss et al., 2021). The data were binarized to represent ion channel presence or absence using a cycle threshold ($C_t$) cutoff (Fig. 2). Several values for the $C_t$ threshold were considered ranging from 13 to 17, but a threshold of 15 cycles was selected so that each neuron included voltage-gated sodium channels and at least one voltage-gated potassium channel to ensure the potential for electrical excitability (Edwards et al., 1995; McAllen et al., 2011). Redundant neurons with the same ion channel combinations were removed to identify unique neuronal genotypes (step II). Corresponding ion channel models were identified from public databases (step III). Fixed conductance values were selected for each (step IV). Known morphological properties of RAGP neurons were incorporated to construct a library of parallel conductance models (step V). Finally, the model responses to the current clamp stimulus were simulated, and the firing properties of each neuronal genotype were analysed and classified (step VI).

Based on thresholding for the apparent presence or absence of the 15 ion channel genes from our thresholded transcriptomic data, we identified 104 unique neuronal genotypes from 321 sampled neurons (Fig. 3A). A maximum of 13 of these ion channels were present in a neuronal genotype. Figure 3A shows these ordered from the most commonly (bottom) to least commonly expressed ion channel gene. Three of the Ca$^{2+}$ channel types (*Cacna1g*, *Cacna1a*, *Cacna1d*) and two of the *Hcn* types (1, 3) were rarely present. The frequency of each neuronal genotype was non-uniform (Fig. 3A). Fifty-seven neuronal genotypes occurred only once; the most common neuronal genotype occurred 22 times. Forty-three percent (137/321) of neurons belonged to eight common types with 14 or more cells in each neuronal genotype: T4, T7, T14, T15, T23, T30, T73 and T91. While only 79/312 neurons expressed *Hcn1*, the commonly occurring types typically had the *Hcn1* channel. Trends in *Hcn2*, *Hcn3* and *Hcn4* were not associated with large increases in the number of occurrences of a neuronal genotype. We also examined sex-based differences in types and whether they projected to the sinoatrial node (SAN). Three of the eight common types were from females, and T30 was found only in female, non-SAN-projecting neurons. There were no statistically significant sex-dependent differences in ion channel expression between the SAN-projecting and non-SAN-projecting RAGP neurons (Moss et al., 2021).

Varying $C_t$ threshold from 13 to 17 caused new neuronal genotypes to appear (Fig. 3B and C). Neuronal genotypes 23 and 91 had 10 or more occurrences for four of five $C_t$ thresholds and neuronal genotypes 4, 30 and 73 were common at three of five $C_t$ thresholds. These results suggest that the majority of cells have common neuronal genotypes that are robust to $C_t$ threshold changes. The percentage of new neuronal phenotypes for $C_t$ thresholds 13, 14, 16 and 17 is 53%, 47%, 56% and 77%. While this is a high percentage of the neuronal phenotypes, further analysis of the frequency of these new neuronal

phenotypes revealed that they account for only 11–20% of the cells.

Responses to current clamp for all neuronal genotypes matched experimentally observed patterns found in multiple species (rodents, minipigs, dogs): phasic responses and tonic firing (Fig. 5) (Armour, 1991; Edwards et al., 1995; Hanna et al., 2021; Harper & Adams, 2021; McAllen et al., 2011). As in experiments, electrophysiological responses were dependent on current clamp stimulus strength, such that some neurons would transition with increased input current (Fig. 5*A*): 61% were phasic only (Fig. 5*D*), 24% tonic only (Fig. 5*E*), 11% phasic-to-tonic (Fig. 5*F*), and 3% tonic-to-phasic (Fig. 5*G*). The remaining firing patterns included stray occurrences of the artificial firing patterns (spontaneous firing, incomplete repolarization), which were filtered out and removed from our analysis. Varying the cycle threshold from 13 to 17 also did not significantly alter the distribution of electrophysiological behaviour for $C_t$ thresholds of 13–17 (Fig. 5*B* and *C*). Phasic behaviour remained dominant. An increase in the relative amount of phasic-tonic firing compared to neurons with tonic firing was observed for $C_t \leq 13$. The primarily phasic and tonic firing patterns emerge from diverse combinations of ion channels that contribute to different dynamics in neuronal behaviours.

The contributions of *Kcnc1* to tonic behaviour were supported by a sensitivity analysis performed for a fixed current stimulus. Firing frequency was observed to be most sensitive to *Kcnc1* and *Cacna1g*. AP peak was most sensitive to *Cacna1a* and inactivating *Kcna1*, while *Cacna1c* and *Kcnj3* affected the maximum hyperpolarization. The full width at half maximum was most regulated by *Cacna1b*. Each metric was most significantly affected by a different ion channel, highlighting the strength of using single-cell transcriptomics to identify the combinations of ion channels present *in vivo*.

To assess the sensitivity of the kinetic parameters of the Na$^+$ channel, we varied the inactivation parameter at steady-state ($h_\infty$) by ±20% (Fig. 6). Increases in the slope shifted the distribution of electrophysiological behaviour so that phasic–tonic and tonic behaviours were predominant. Decreases in the slope maintained the predominance of phasic behaviour. Of the alternative Na$^+$ channel models considered, two models had similar $h_\infty$ values while one model had an inactivation parameter two times higher (Table 1). Thus, the uncertainty of kinetic parameters in ion channel models should be considered when analysing the model predictions.

Current–frequency curves for tonically active neuronal models demonstrated a monotonic increase in frequency with increased current for all tonically-firing neurons (Fig. 7). The slopes of the *f–I* curves were clustered into two groups. The neuronal genotypes which expressed either both *Cacna1d* and *Kcna1+ab1* (inactivating *Kcna1*) or neither, had slopes greater than the best-fit line. The neuronal genotypes which expressed *Cacna1d* but lacked *Kcna1+ab1* had slopes less than the best-fit line. Despite these differences, the firing frequencies of our neuronal models ranged between 12 and 65 Hz for a stimulus range of 0.01–0.5 nA, which is within the experimentally observed limits of 5–60 Hz (guinea pigs: 5 Hz; rats: 9–15 Hz; mice: 2–60 Hz; dogs: 60 Hz; and humans 20–50 Hz) (Edwards et al., 1995; McAllen et al., 2011; Tompkins et al., 2025).

To check the model's relationship with the original transcriptomic data from which it was derived, the relative expression levels of the ion channels to the maximal conductance values identified during model development were compared (Fig. 8). Conductance values and expression levels were calculated as fold difference from *Cacna1a*, which had the lowest conductance value. The model-predicted ratio of conductance values was correlated with the ratio of expression levels in the transcriptomic data ($R^2 = 0.67$). The average relative expression and conductance values for the ion channels that are critical for firing behaviour, *Scn1a* and *Kcna1*, also correspond (3.4 and 4.2, respectively). There are some exceptions to the correlation, notably *Hcn3* and *Cacna1b*, which were removed for the correlation analysis. It should be noted that the maximal conductance values were selected based on the passive electrical properties of the neurons, not the transcriptomic data. Taken together, these findings suggest that the model predicted conductance values independently correlate with the relative expression of ion channels.

## Discussion

### Computational systems biology approach augments electrophysiology data

Biophysical models have long been limited by the electrophysiology data available, sparse in most species due to the time and skill required to collect it, and almost entirely unavailable for humans. Sample sizes for electrophysiology datasets are also limited, while transcriptomic data can be gathered more quickly and obtained in quantities of hundreds to thousands of single cells (Moss et al., 2021; Zhao et al., 2022). Therefore, transcriptomics coupled with a computational systems biology approach can be used as a complementary resource that will allow more rapid development of extensive cell activity libraries (McDougal et al., 2017; Podlaski et al., 2017).

We aimed to connect single-neuron transcriptomic data to cellular electrophysiology with quantitatively validated electrophysiological models. Our model is the latest iteration of our work to understand ion channel contributions to electrophysiological behaviour and cardiovascular regulation (Schwaber et al., 1993;

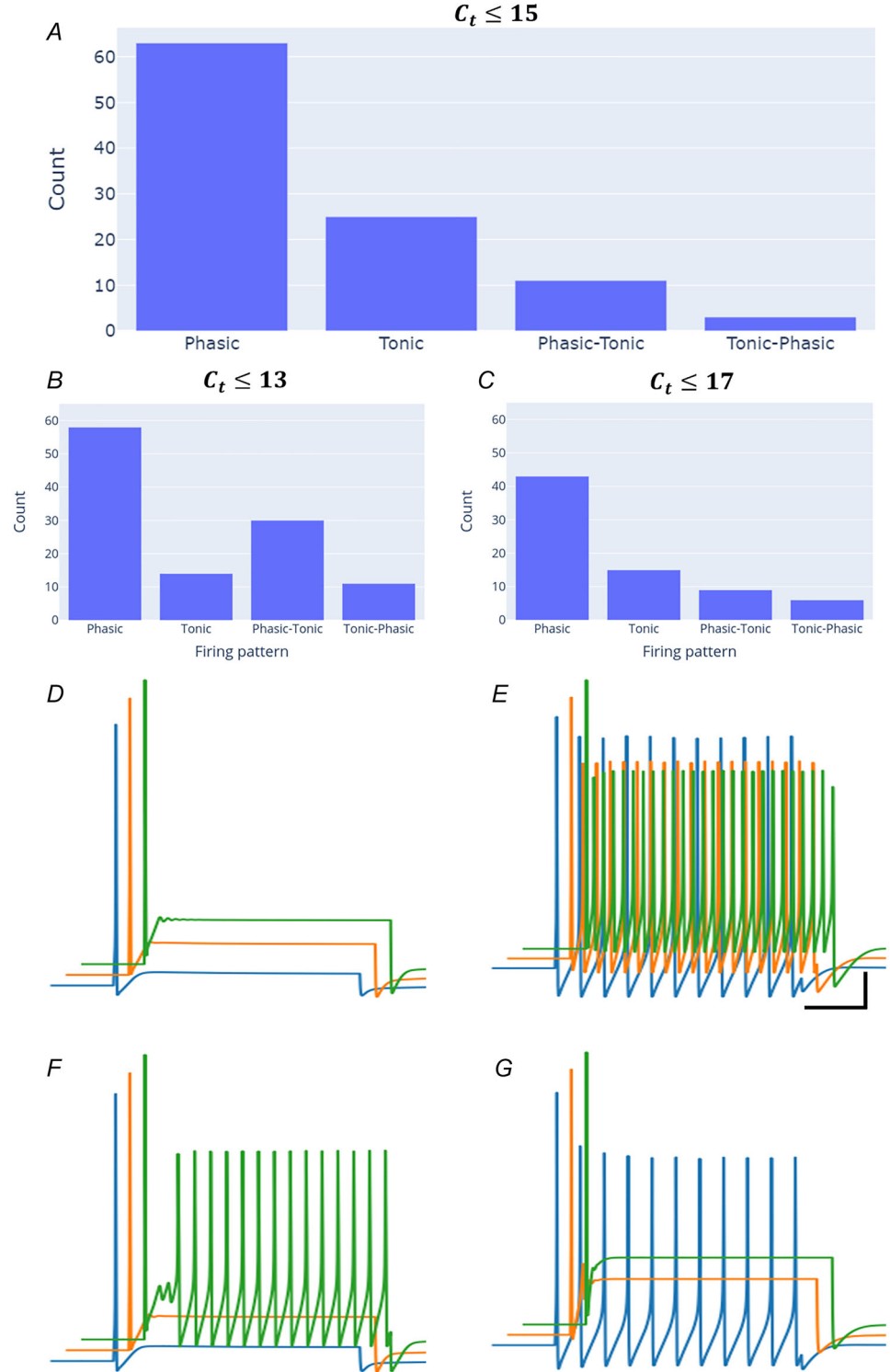

**Figure 5. Neuronal firing behaviour with increasing stimulus strength**

*A*, number of neurons with each firing behaviour for $C_t \leq 15$. *B* and *C*, firing pattern distribution for $C_t \leq 15$ (*B*) and $C_t \leq 15$ (*C*). *D–G*, example traces for phasic (T1) (*D*), tonic (T51) (*E*), phasic-to-tonic (T2) (*F*) and tonic-to-phasic (T52) (*G*). Scale: 10 mV, 100 ms. Blue, orange, green: 0.1, 0.3, 0.5 nA stimulus. [Colour figure can be viewed at wileyonlinelibrary.com]

Vadigepalli et al., 2001), made possible by the increasing availability of single-cell transcriptomic data. Despite limitations due to gaps in datasets containing both transcriptomic data and electrophysiological recordings in ICNS neurons, as are available for other neuronal classes in the brain (Bernaerts et al., 2023; Nandi et al., 2022), we developed a library of biophysically constrained ICNS parallel conductance models. In this paper, we propose an alternative modelling approach, built on the basis of the more widely available and fine-grained transcriptomic data to create electrophysiological models that match the distribution of behaviour observed in populations of similar neurons (Edwards et al., 1995; McAllen et al., 2011; Tompkins et al., 2025). The advantage of our approach is that the low throughput and laborious process of collecting electrophysiology data can be augmented by combining it with computational approaches to effectively increase the sample size.

## Distribution of tonic and phasic firing patterns

Our model yielded 61% phasic, 24% tonic, 11% phasic-to-tonic, 3% tonic-to-phasic and ~1% artificial firing patterns (spontaneous firing, incomplete repolarization). Our model parameters were additionally tuned to ensure that the model does not generate firing patterns that are artifacts of the modelling paradigm (see Methods). Despite generating physiologically observable patterns, the presence of these trace firing patterns leads us to hypothesize that either the transcriptomic sequences of ion channels in those respective neuronal genotypes are inadequately identified, or these neuronal types may have been erroneously identified as a RAGP PN, whereas it may be a non-excitable neuron. Thus, the strength of our workflow can be additionally employed to enhance experimental protocols and insights.

Some of our models transitioned from phasic to tonic behaviour with increasing stimulus strength (Fig. 5*F* and *G*). Some neurons remain phasic (Fig. 5*D*) or tonic (Fig. 5*E*) over the same stimulus range. In our set of 104 neuronal genotypes, 40/104 consistently exhibited a robust phasic firing profile while, the remaining 64/104 elicited phasic and tonic firing patterns. This finding lends itself to the qualitative segregation of our neuronal models on the basis of their stability with respect to parametric variations within physiological ranges. The

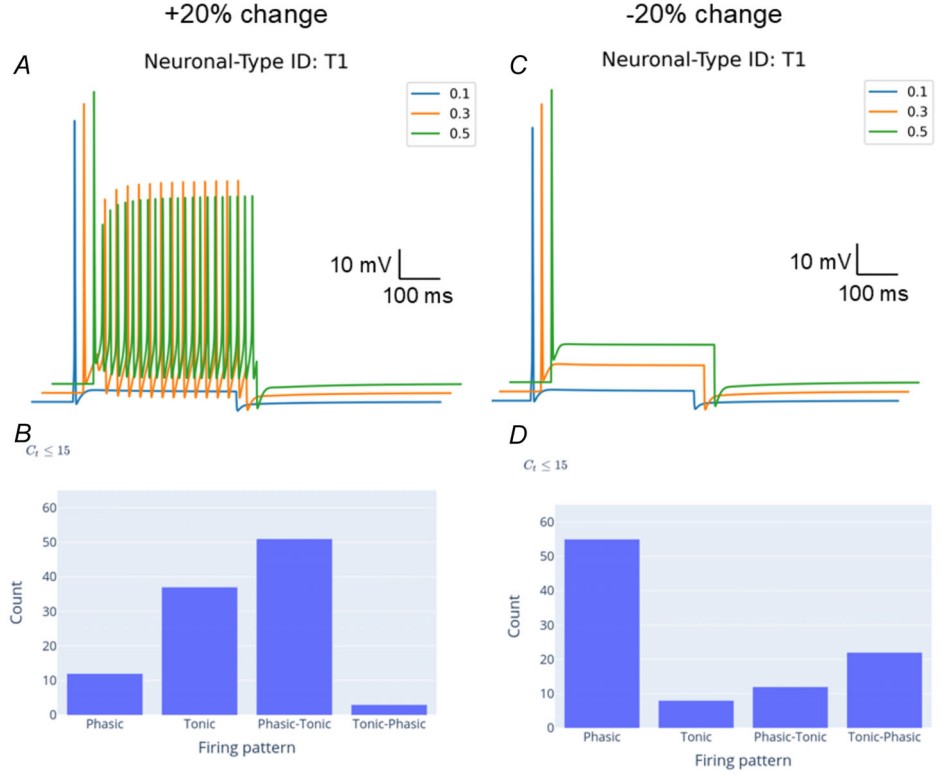

**Figure 6. Effect of varying the Nav1.1 channel inactivation parameter ($h_\infty$) on electrophysiological behaviour**
Simulations are shown for a +20% change in $h_\infty$ (*A* and *B*) and a −20% change in $h_\infty$ (*C* and *D*). A $C_t$ threshold of 15 was used so that results for the electrophysiological behaviour of neuronal genotype T1 can be compared to the results in *D*, which shows that T1 has phasic behaviour at all three stimulus intensities. $h_\infty$ values varied within the range of what would be expected for inter-model variability. [Colour figure can be viewed at wileyonlinelibrary.com]

dominant firing behaviour in our models was phasic over a range of 0.1–0.5 nA current clamp stimulus. Minipig, dog and rat ICNS neurons show dominantly phasic responses to current clamp (McAllen et al., 2011; Tompkins et al., 2025; Xi et al., 1994), consistent with our model, which was based on Yucatán minipig transcriptomic data. Different studies on guinea pig neurons have reported them to be

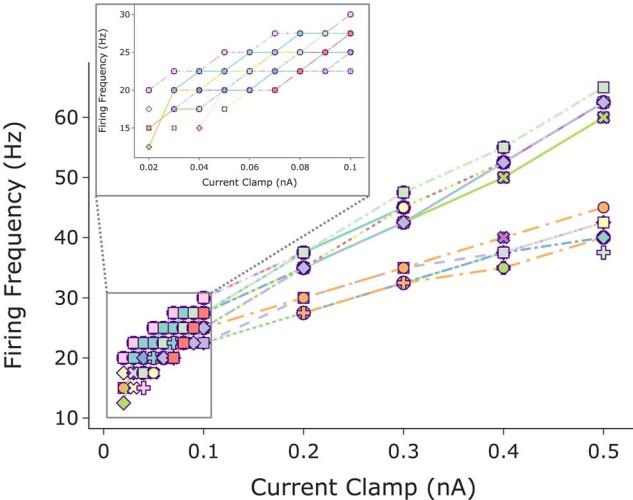

**Figure 7. Current–frequency relationship for tonically firing neurons**
Data fit (black line) with slope 60.5 Hz/nA. Each colour represents a different neuronal genotype. [Colour figure can be viewed at wileyonlinelibrary.com]

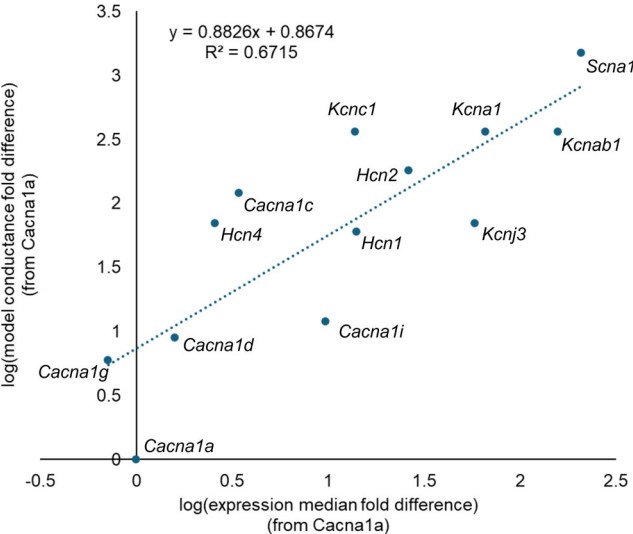

**Figure 8. Relative expression ratios correlate with maximal conductances identified in the model**
Conductance and expression ratios are relative to *Cacna1a*, which had the lowest conductance in the model. Expression fold differences relative to *Cacna1a* were log transformed before $R^2$ was calculated. The outliers *Hcn3* and *Cacna1b* are not shown and were excluded from the correlation analysis. [Colour figure can be viewed at wileyonlinelibrary.com]

either phasic (Hoover et al., 2009) or tonic (Edwards et al., 1995). By contrast, human neurons show predominantly tonic responses (Edwards et al., 1995; Tompkins et al., 2025). It will be interesting to see if transcriptomic differences across species can be used to predict these electrophysiological differences.

## Strengths and limitations of the model

A significant challenge for our approach remains the unknown effects of several steps between mRNA transcription and placement of ion channels in the membrane: post-transcription processing, translation and post-translational processing, subunit aggregation, and differential channel placement in dendrites, soma and axons. Owing to these factors, the direct use of cycle threshold as an indicator of ion channel density was not successful. Instead, we employed a novel cutoff technique to binarize transcriptomic data into ion channel presence or absence in a neuron if the cycle threshold was above or below 15. This technique was effective for the model given that *post hoc* analysis found that model-predicted relative conductance correlated with the relative expression found in the transcriptomic data.

Proteomic data would be closer to the physiological product and, therefore, should be more useful in improving omic-to-model translation by providing a one-to-one match between the genetic and electrophysiological profiles. Alternatively, patch-clamp data, alongside transcriptomic data and electrophysiology data from the same cell could further enhance model translation (Bernaerts et al., 2023). Patch-clamp has been used to generate models that demonstrate predicted conductances that reflect gene expression differences (Nandi et al., 2022). However, patch-clamp data use is still limited by its low throughput. The limited amount of protein expression data available on the relative expression of ion channels shows that the model-predicted relative conductances are reasonable. For example, Nav1.1 expression is about 1.4 times greater than Kv1.1 expression in mammalian central neurons (Gu et al., 2018). The translation of ion channel expression to conductance is further convoluted by differences in single channel conductance between ion channels. Single channel conductance is approximately 1.5-fold higher for Nav1.1 (17 pS) compared to Kv1.1 (12 pS) (Streit et al., 2014; Vanoye et al., 2006). The combination of 1.4-fold expression and 1.5-fold channel conductance differences results in a 2-fold higher conductance for Nav1.1 compared to Kv1.1, which is comparable to 4.2-fold higher model-predicted conductance.

A further limitation in the use of transcriptomic data arose because of limitations in the ion channel models found in the available databases. The models were selected from three public databases: Channelpedia (Ranjan et al.,

2011), ModelDB (McDougal et al., 2017) and Ion Channel Genealogy (Podlaski et al., 2017). Channelpedia, an initiative by the Blue Brain Project to develop ion channel models from in-house electrophysiological experiments, was our preferred database for ion channel models. However, the models in Channelpedia described the biophysics of the respective homomeric ion channels. Physiologically, heteromeric forms of these ion channels exist where the heteromers impart different biophysics to the channel operation. In our case of Kv 1.1, the presence of *Kcnab1* ($\beta$1 subunit of Kv 1.1) was accounted for by introducing an inactivation variable to the published KDR model. Owing to this multitude of ion channel behaviour, we additionally employed those models from ModelDB whose provenance could be established by Ion Channel Genealogy. In cases where multiple ion channel models from these databases appeared equally probable to describe ion channel behaviour, we shortlisted them by comparing their gating kinetics (Fig. 4). However, we were limited by the rigour in the ion channel kinetic models available to us.

A potential limitation is perhaps our choice of the Hodgkin–Huxley formalism of ion channel models. We propose a novel methodology to develop biophysically constrained models informed by transcriptomics. Since the inclusion of a gene-based ion channel model was crucial for its demonstration, we did not explore the applicability of gene-independent Izhikevich and FitzHugh–Nagumo spiking neuron models. Alternatively, unlike Hodgkin–Huxley models, we did not find similar gene-specific multi-state Markov kinetic models for the identified ion channel genes. Although the Hodgkin–Huxley model continues to remain a gold standard for ion channel modelling, it is restricted by the identifiability of its parameters (Meunier & Segev, 2002; Walch & Eisenberg, 2016). The 1952 Hodgkin–Huxley model assigned the gating powers by careful curve-fitting to the ionic current traces. In order to circumvent this apparent shortcoming, Channelpedia was our preferred database. Since Channelpedia kinetic models are based on similar experiments on singularly expressed homomeric ion channels, the gating powers assigned to their models are likely to be backed by a defensible rigour. There has been a continued effort to bridge the gap between model parameters and ion channel dynamics by either determining the gating dynamics from single channel currents (Sigg, 2014; Strassberg & DeFelice, 1993), by ascertaining conditions under which the gating parameters are identifiable (Walch & Eisenberg, 2016), or by leveraging neural networks (personal communication, B. Prokop: Prokop et al., 2024). However, addressing the strength of the gating dynamics in already published models is beyond the scope of our current work. Our proposed methodology is invariant to the formalism of ion channel models that are linked to specific genes,

and thus any rigorously developed model with known gene composition may be explored in the integrated representation of neuronal electrical behaviour.

The ion channel models were derived either from large cells such as mammalian cortical pyramidal cells or from gene expression in oocytes, human embryonic kidney (HEK) cells, or other heterologous cell types (McDougal et al., 2017; Podlaski et al., 2017; Ranjan et al., 2011), any of which will have dynamics different from that of similar channels from RAGP neurons. In particular, we encountered this difficulty with the Na$^+$ channel, a channel that is particularly difficult to measure in voltage clamps due to fast kinetics. Multiple isomers of this channel distributed along dendrites and axons of different neurons, possess different kinetics. Because our modelling approach utilizes transcriptomic data, this model more specifically identifies the ion channels used compared to prior parallel conductance models that only specify current types such as the potassium delayed rectifier which can be attributed to multiple ion channel genes (Rybak et al., 1997; Yaghini Bonabi et al., 2014). Both electrophysiological recordings and modelling of single isolated neurons are also limited by the artificiality of current-clamp inputs. The use of spike train inputs based on vagal recordings and cellular recordings from intact ganglion preparations will allow further development and refinement of computational models (Machhada et al., 2015; Rentero et al., 2002).

Several planned extensions of the current model will address some of these limitations. The most immediate extension is to link the single-neuron models into an ICNS network model. One way to do this would be to add synapse models based on electrophysiology data (McAllen et al., 2011) and connect the neurons based on neural tracing data (Cheng et al., 2004). The implementation of spike train stimuli from vagal recordings could then be used as inputs to the model to test it against ICNS electrophysiology (Machhada et al., 2015; Rentero et al., 2002). In addition, there have been instances where a parameter shift in a single-neuronal Hodgkin–Huxley model has resulted in changed firing characteristics (Doi & Kumagai, 2005; Guckenheimer & Labouriau, 1993; Izhikevich, 2003; Postnova et al., 2007; Rush & Rinzel, 1995). However, the firing characteristics of single-neurons are rapidly adjusted when in a network. *In vivo*, the PN forms a network within the RAGP wherein the tonic, phasic-to-tonic and tonic-to-phasic neuronal genotypes (Fig. 5*D–G*) will interconnect with the robustly phasic neurons, and non-excitable small intensely fluorescent cells (Hanna et al., 2021; Moss et al., 2021). The individual firing characteristics of these single neurons will thereby be mediated by the strength of their connections as well as the inputs received from the vagus nerve. While the strength of the couplings does not apply for single-neuron models (Nowotny & Rabinovich, 2007;

Postnova et al., 2007), a bifurcation analysis framework may be an effective analysis strategy as we extend this work to examine our PN models in a network.

Our model is also primed to be coupled with other cardiovascular physiology models to address questions on ICNS regulation of heart function (Gee, Lenhoff, et al., 2023; Park et al., 2020). In the context of ICNS function in cardiovascular regulation, we hypothesize that the neurons predisposed towards phasic and tonic behaviour may have different functions. Due to their higher firing rate, the tonic neural activity may carry beat-to-beat tone from the nucleus ambiguus (NA) via the vagus to regulate heart rate, while physically firing neurons may regulate contractility with a slower drive from the dorsal motor nucleus of the vagus (DMV). This is supported in part by neural tracing studies, which showed the NA and DMV project to distinct populations of PN within the same ICNS ganglion, extending the two separate lanes of vagal tone from the brainstem to the ICNS (Cheng et al., 2004; Gee, Hornung, et al., 2023). Physiological evidence also supports the notion of two distinct vagal lanes, as different ICNS neuronal clusters, such as the RAGP, have been found to primarily regulate heart rate versus contractility (Fedele & Brand, 2020; Gourine et al., 2016). Our model, in the present state, could be incorporated as a module in a larger systemic model of autonomic regulation and cardiovascular function (Gee, Hornung, et al., 2023; Park et al., 2020) to test the functional implications of this hypothesized connectivity.

## Conclusion

We demonstrate a novel workflow to bridge the gap between molecular-level gene expression and cellular-level electrophysiological function, using single-neuron HT-qPCR data from RAGP neurons to derive ion channel combinations that generate a library of single-neuron parallel conductance models (Fig. 1). In order to develop biophysically detailed models from single-cell transcriptomic data, we used single-neuron HT-qPCR ion channel data from RAGP PN to quantitatively derive ion channel combinations that generate a library of single-neuron parallel conductance models. We used a gene expression threshold to select the presence or absence of ion channels in each parallel conductance model. After thresholding, 104 unique ion channel combinations were identified from the 321 single-neuron transcriptomic samples (Fig. 3). The emergent firing patterns were in agreement with experimental reports (Fig. 5) and the model-predicted conductance ratios correlated with the expression ratios in the transcriptomic data (Fig. 8). By this approach, we demonstrate a use case of computational modelling to relate molecular data to the electrical behaviour

of neurons. This library of transcriptomics-based single-neuron models provides a framework for developing parallel conductance models of neurons from other regions and provides a platform for developing network models that represent the interactions of various neuronal genotypes involved in cardiovascular regulation.

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

## Additional information

### Open research badges

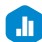

This article has earned an Open Data badge for making publicly available the digitally-shareable data necessary to reproduce the reported results. The data is available at https://modeldb.science/2018157.

### Data availability statement

The model source and analysis code are available on GitHub (https://github.com/Daniel-Baugh-Institute/BiophysicalModellingOfIntrinsicCardiacNeurons) and the SPARC portal (https://sparc.science/,10.26275/cy9w-ttjn). Model source code is also available through ModelDB (https://modeldb.science/2018157). The model can also be accessed through the simulation platform oSPARC (https://osparc.io/), which enables users to run simulations with the model using a Graphical User Interface. The Ten Simple Rules for the Credible Practice of Modelling and Simulation in Healthcare (Erdemir et al., 2020; Mulugeta et al., 2018) were documented to ensure data availability and model reproducibility and can be found on GitHub.

### Competing interests

The authors declare that the research was conducted in the absence of any commercial or financial relationships that could be construed as a potential conflict of interest.

### Author contributions

W.W.L., R.V. and J.S.S, conceived, supervised the work and provided significant editorial contributions. S.G. worked on model development, validation and interpretation of data. A.J.H.N and W.W.L. handled NetPyNE software development, designing and implementation of the Sobol algorithm. S.G. and L.K. worked on building the model and ion channel library. S.G. and M.M.G. worked on the analysis and visualization of data. M.M.G. and A.M. assisted in the interpretation of transcriptomic data. J.D.T shared experimental inputs to assess model performance. S.G. and M.M.G. wrote the manuscript with edits from J.D.T., J.S.S., R.V. and W.W.L. All authors provided critical feedback to improve the manuscript. All authors have read and approved the final version of this manuscript and agree to be accountable for all aspects of the work in ensuring that questions related to the accuracy or integrity of any part of the work are appropriately investigated and resolved. All persons designated as authors qualify for authorship, and all those who qualify for authorship are listed.

### Funding

National Heart, Lung, and Blood Institute (NHLBI) U01 HL133360 and R01 HL161696: J.S., R.V.; National Institutes of Health (NIH) Common Fund Stimulating Peripheral Activity to Relieve Conditions (SPARC) Program OT2OD030534: R.V.; National Science Foundation (NSF) 1940700: M.G.

### Acknowledgements

We would like to express our gratitude to Professor Robin McAllen for his insights and inputs which aided our model development and validation. We would also like to thank Professor Babatunde A. Ogunnaike for his mentorship. We are grateful to Siyan Guo and Joy Wang for reproducing current clamp simulations for neuron T54 from an earlier draft of this manuscript, and to Dr Sujata Patil for reproducing the figures from the analysis code as part of 10 rules for Credible Practice of Modelling and Simulation in Biomedicine. This research was supported in part through the use of DARWIN computing system: DARWIN – A Resource for Computational and Data-intensive Research at the University of Delaware and in the Delaware Region, Rudolf Eigenmann, Benjamin E. Bagozzi, Arthi Jayaraman, William Totten, and Cathy H. Wu, University of Delaware, 2021, URL: https://udspace.udel.edu/handle/19716/29071.

### Keywords

autonomic regulation, Hodgkin–Huxley model, intrinsic cardiac nervous system, ion channels, systems biology, vagus nerve

## Supporting information

Additional supporting information can be found online in the Supporting Information section at the end of the HTML view of the article. Supporting information files available:

**Peer Review History**

