## [Peer Review History · The Journal of Physiology]

Biophysical Modelling of Intrinsic Cardiac Nervous System Neuronal Electrophysiology based on Single-cell Transcriptomics

Suranjana Gupta, Michelle M Gee, Adam John Hunter Newton, Lakshmi Kuttippurathu, Alison Moss, John D. Tompkins, James S Schwaber, Rajanikanth Vadigepalli, and William Lytton

DOI: 10.1113/JP287595

Corresponding author(s): Rajanikanth Vadigepalli (Rajanikanth.Vadigepalli@jefferson.edu)

Review Timeline:

Submission Date:	30-Aug-2024
Editorial Decision:	12-Nov-2024
Revision Received:	31-Dec-2024
Accepted:	14-Feb-2025

Senior Editor: Natalia Trayanova

Reviewing Editor: Eleonora Grandi

Transaction Report:

Dear Dr Vadigepalli,

Re: JP-RP-2024-287595 "Biophysical Modelling of Intrinsic Cardiac Nervous System Neuronal Electrophysiology based on Single-cell Transcriptomics" by Suranjana Gupta, Michelle M Gee, Adam Newton, Lakshmi Kuttippurathu, Alison Moss, John D. Tompkins, James S Schwaber, Rajanikanth Vadigepalli, and William Lytton

Thank you for submitting your manuscript to The Journal of Physiology. It has been assessed by a Reviewing Editor and by 1 expert referees and we are pleased to tell you that it is potentially acceptable for publication following satisfactory major revision.

REVISION CHECKLIST:

We look forward to receiving your revised submission.

Yours sincerely,

Natalia Trayanova
Senior Editor
The Journal of Physiology

Reviewing Editor's comments:

Apologies for the late review. One reviewer did not return their assessment and stopped responding to our email.

The reviewer finds the aim of this study to be of interest, but points out a number of concerns regarding the approach - including the model of choice for this study. The authors are also invited to provide experimental evidence for their assumptions or accepted parameter combinations. Please note a marked up PDF is available in addition to the summary or critiques.

Referee #2:

The study is motivated by a fascinating idea, which is, being able to explicitly include information about patterns of ion channel expression into an excitable membrane model, in this case, from intrinsic cardiac neuron transcriptomics. However, the models were chosen from databases that assume channel dynamics, in particular gating dynamics, that by their formulation bias the predictions in at least a couple of ways. The most important one is the use of powers in the gating terms, which impact the sizes of the maximal conductances of the model. One key question that arises from this is: are the ratios of Na and K maximal conductances equivalent to the ratios of the levels of expression from the data corresponding to Na and K channels? The models chosen for the study, as presented, predict ratios of 3-5 times as many Nav1.1 channels as

Kv1.1 channels, for instance. Is this accurate? I would very much like to see some evidence to that effect. Electrophysiological recordings and other splice variant data indicate the opposite (several fold more K channels in comparison to Na channels, for instance), but that is not conclusive data that applies to all species and neurons within them.

Regarding originality, the question has been asked before (e.g. Herrera-Valdez, et al. 2013, Nowotny et al. 2007), but to my knowledge, there have not been any studies in which transcriptomic or proteomic data have been used. In this regard, the study is motivated by a very interesting paradigm linking modelling and experimental measurements, but again, choose arbitrarily from model databases arguing that many people use those models (see comments about powers in gating terms in annotated pdfs and above). One important fact to consider is that the behaviour of the neurons of interest in the study can be captured initially with a simple 2D model with voltage-gated Na and K-delayed rectifier channels. The conclusions from using a model with high dimensionality are hard to interpret and it is easy to conclude, wrongly, that "wild combinations of parameters indicate wild combinations of expression levels", as suggested by some. Highly dimensional models may show combinations of parameters that may not be correct in light of physiological considerations. Also, the authors choose to model ionic currents with different functional forms (fundamentally different assumptions for the underlying biophysics) within the same model, which does not help for interpretation.

As to the dynamics of the model, since there seems to be a gradient of behaviours that stem all from phenomena captured by 2D biophysics, I think the authors could have focused on building up from simple models to tackle the issues of diversity in the neuronal population, to network dynamics as mentioned in the discussion of the paper. The model does not go significantly further in making predictions, in comparison to the transcriptomics data, but analysing the dynamics from a tractable perspective would have been more informative, and then the inclusion of more ion channels to explain how redundancies occur would have been appropriate.

In spite of the poor modelling and choices, and having done no geometrical or other kinds of analysis that are more meaningful than the statistics, I believe that the approach heads fundamentally in the right direction, and other modelling approaches could complement what is reported in this study, provided that the transcriptomic data is shared properly (not just the binary data about whether a channel gene is transcribed or not).

END OF COMMENTS

The Journal of Physiology

<https://jp.msubmit.net>

JP-RP-2024-287595

Title: Biophysical Modelling of Intrinsic Cardiac Nervous System Neuronal Electrophysiology based on Single-cell Transcriptomics

Authors: Suranjana Gupta
Michelle Gee
Adam Newton
Lakshmi Kuttippurathu
Alison Moss
John Tompkins
James Schwaber
Rajanikanth Vadigepalli
William Lytton

Author Conflict: No competing interests declared

Author Contribution: Suranjana Gupta: Acquisition, analysis or interpretation of data for the work; Drafting the work or revising it critically for important intellectual content; Final approval of the version to be published; Agreement to be accountable for all aspects of the work Michelle Gee: Acquisition, analysis or interpretation of data for the work; Drafting the work or revising it critically for important intellectual content; Final approval of the version to be published; Agreement to be accountable for all aspects of the work Adam Newton: Acquisition, analysis or

Disclaimer: This is a confidential document.

interpretation of data for the work; Drafting the work or revising it critically for important intellectual content; Final approval of the version to be published; Agreement to be accountable for all aspects of the work
Lakshmi Kuttippurathu: Acquisition, analysis or interpretation of data for the work; Drafting the work or revising it critically for important intellectual content; Final approval of the version to be published; Agreement to be accountable for all aspects of the work
Alison Moss: Acquisition, analysis or interpretation of data for the work; Drafting the work or revising it critically for important intellectual content; Final approval of the version to be published; Agreement to be accountable for all aspects of the work
John Tompkins: Acquisition, analysis or interpretation of data for the work; Drafting the work or revising it critically for important intellectual content; Final approval of the version to be published; Agreement to be accountable for all aspects of the work
James Schwaber: Conception or design of the work; Drafting the work or revising it critically for important intellectual content; Final approval of the version to be published; Agreement to be accountable for all aspects of the work
Rajanikanth Vadigepalli: Conception or design of the work; Acquisition, analysis or interpretation of data for the work; Drafting the work or revising it critically for important intellectual content; Final approval of the version to be published; Agreement to be accountable for all aspects of the work
William Lytton: Acquisition, analysis or interpretation of data for the work; Drafting the work or revising it critically for important intellectual content; Final approval of the version to be published; Agreement to be accountable for all aspects of the work

Running Title: Biophysical Modeling of Intrinsic Cardiac Neurons

Dual Publication: No

Funding: HHS | NIH | National Heart, Lung, and Blood Institute (NHLBI): James S Schwaber, Rajanikanth Vadigepalli, U01 HL133360; HHS | NIH | National Heart, Lung, and Blood Institute (NHLBI): James S Schwaber, Rajanikanth Vadigepalli, R01 HL161696; HHS | NIH | NIH Office of the Director (OD): Rajanikanth Vadigepalli, OT2 OD030534

**Biophysical Modelling of Intrinsic Cardiac Nervous System Neuronal**
**Electrophysiology based on Single-cell Transcriptomics**

Suranjana Gupta^{1,†}, Michelle M. Gee^{2,3,†}, Adam J.H. Newton¹, Lakshmi
Kuttippurathu², Alison Moss², John D. Tompkins⁴, James S. Schwaber^{2,3},
Rajanikanth Vadigepalli^{2,3,*}, and William W. Lytton^{1,5}

¹*Department of Physiology and Pharmacology, SUNY Downstate Health Sciences*
*University, Brooklyn, NY, USA*

²*Daniel Baugh Institute for Functional Genomics/Computational Biology, Department of*
*Pathology and Genomic Medicine, Thomas Jefferson University, Philadelphia, PA, USA*

³*Department of Chemical and Biomolecular Engineering, University of Delaware, Newark,*
*DE, USA*

⁴*Department of Medicine, Cardiac Arrhythmia Center and Neurocardiology Research*
*Program of Excellence, University of California, Los Angeles, CA, USA*

⁵*Department of Neurology, Kings County Hospital, Brooklyn, NY, US*

**Corresponding author: Rajanikanth Vadigepalli, Rajanikanth.Vadigepalli@jefferson.edu*

*†Co-first author*

Abstract

The intrinsic cardiac nervous system (ICNS), termed as the heart's "little brain", is
the final point of neural regulation of cardiac function. Studying the dynamic
behaviour of these ICNS neurons via multiscale neuronal computer models has been
limited by the sparsity of electrophysiological data. We developed and analysed a
computational library of neuronal electrophysiological models based on single
neuron transcriptomics data obtained from ICNS neurons. Each neuronal phenotype
was characterized by a unique combination of ion channels identified from the
transcriptomic data, using a cycle threshold cutoff that ensured electrical excitability
of the neuronal models. **The parameters of the ion channel models were grounded**
**based on passive properties: resting membrane potential, input impedance, and**
**rheobase**. Consistent with experimental observations, the emergent model dynamics
showed phasic activity in response to current clamp stimulus in a majority of
neuronal phenotypes (61%). Additionally, 24% of the ICNS neurons showed tonic
response, 11% were phasic-to-tonic with increasing current stimulation, and 3%
showed tonic-to-phasic behaviour. The computational approach and the library of
models bridge the gap between widely available molecular-level gene expression
and sparse cellular-level electrophysiology for studying the functional role of the
ICNS in cardiac regulation and pathology.

Key points

- • We developed computational models of neuron electrophysiology from single-
cell transcriptomics data from neurons in the heart's "little brain": the intrinsic cardiac
nervous system.
- • **The single-cell transcriptomic data formed the basis for ion channel**
**combinations in each neuronal model.**
- • The library of neuronal models was constrained by the passive electrical
properties of the neurons and predicts a distribution of phasic and tonic responses
that aligns with experimental observations.
- • **Heterogeneity in the electrophysiological behavior of neurons is driven by**
**variation in ion channel expression.**
- • These neuron models are a first step towards connecting single-cell
transcriptomics data to dynamic, predictive physiology-based models.

Introduction

Parasympathetic and sympathetic imbalance contributes to the etiology of many
cardiovascular diseases. A key regulator of sympathovagal balance is the heart's
"little brain", the intrinsic cardiac nervous system (ICNS), which contains both
cholinergic and catecholaminergic neurons (J. A. Armour, 2008; Hadaya & Ardell,
2020; Hanna et al., 2021; Moss et al., 2021). As the final neural regulatory point for
the heart, the ICNS mediates the balance of parasympathetic and sympathetic inputs
to the cardiac tissue. Neural remodelling within ICNS has been linked to the
progression of cardiovascular disease (Beaumont et al., 2016; Salavatian et al.,
2016; Vaseghi et al., 2017). Both phasic and tonic firing responses have been
observed in mouse, pig, and human, and neural remodelling to modulate these
behaviors could regulate sympathovagal balance in health and disease (Tompkins et
al., 2024).

Phasic and tonic electrophysiological behavior arises from a multitude of ion channel
combinations through a complex mapping relating variable molecular expression to
relative more constrained functional responses. Recently, the increased availability
of high-throughput, single-neuron transcriptomics has made it possible to identify the
exact combinations of ion channels present in each cell to connect subcellular
components to cellular function (Moss et al., 2021). These transcriptomic datasets
have been mined to address questions of how ion channel degeneracy contributes to
firing robustness in neurons (Drion et al., 2015; Goillard & Marder, 2021; Nandi et
al., 2022; Roy & Narayanan, 2023), but have not been used to study how high-
dimensional ion channel expression collapses to a restricted set of phasic and tonic
responsive phenotypes. Differences in ion channel conductances that drive
transitions from phasic to tonic firing in a neuron are potential regulatory points for
controlling sympathovagal balance in the ICNS.

To address this question, we use the well-established Hodgkin-Huxley models and
combine them with single-cell transcriptomics data to identify specific ion channel
combinations for each neuron. This computational approach allows us to explore the
contributions of individual ion channels that would not be possible without inferring
channel involvement through time-consuming pharmacological blockades or
assuming channel types (Schwaber et al., 1993; Shevtsova et al., 2020). Instead, *in-*

*silico* screening can be performed to identify the most important ion channels for
further experimental testing. In addition, neurons of the same cell type have
electrophysiological behavior consistent with each other in response to current clamp
stimulus, but vary in their ion channel conductance densities. This heterogeneity may
contribute to the variable responses of neurons of the same type to perturbations,
muddling the association between an ion channel and a particular function identified
via conventional experimental approaches (Goaillard & Marder, 2021).

Electrophysiological recording of neuronal electrical activity has been a productive
approach to studying the ICNS to capture neuronal firing rate and membrane
electrical behavior, but it is labor-intensive and, therefore, low throughput. More
recently, the systems biology approach provides a complementary approach by
capitalizing on high-throughput transcriptomic techniques (Hanna et al., 2021; Moss
et al., 2021) that are becoming increasingly available through data sharing initiatives,
such as the National Institutes of Health's SPARC program (<https://sparc.science/>).

In this work, we aim to connect the electrophysiological behavior of ICNS neurons to
their gene expression using transcriptomics-based single-cell parallel conductance
Hodgkin-Huxley neuronal phenotype computational models. We present a strategy
for using single-neuron transcriptomic data to predict neuronal membrane
physiology, demonstrating a workflow for building a library of neuronal phenotype
models. We used data from 321 porcine RAGP neurons to deduce the presence or
absence of particular channel types in each neuron. We then used Hodgkin-Huxley
ion-channel models from open-source model repositories to construct a library of
parallel-conductance models reflecting ion channel combinations and predicting
electrophysiological behavior.

**Materials and Methods**

We propose a 6-step workflow for the development of electrophysiological neuronal
models from single-neuron transcriptomics data (Figure 1). We expand upon Steps I
and V in Section 2.1, Step III in Section 2.2, and Step IV in Section 2.3.

**Morphology, physiology, and transcriptomics of neurons**

[revised manuscript text omitted]

Channel conductances were primarily constrained by the passive electrical
properties of the neurons. We had an additional degree of freedom in the fractional
amplitude parameter (ϕ_{Kcnc1}), which represents the relative contributions of the fast
and slow activation processes in the *Kcnc1* channel. We observed this parameter to
mainly control the firing rates (Figure S4). We varied ϕ_{Kcnc1} over a range of stimulus
currents, and settled on a value that restricted the firing rates in accordance with
experimental data (Harper & Adams, 2021; Vaseghi et al., 2017).

Simulations were run using the three degenerate parameter sets identified.
Additional tuning of the channel conductances were required in cases where
ensemble models were dominated by artificial firing patterns. These patterns
included continued firing without the application of a stimulus, sustained firing activity

post removal of the stimulus and incomplete repolarization resulting in elevated RMP
 for prolonged periods of time.

**Table 1: Model parameters from literature**

Parameter	Value	Parameter	Value
Simulation Time	1000 ms	Time Step (dt)	25 μ s
V_{rmp}	-61 mV	Membrane Threshold	-10 mV
Soma Length	21 μ m	Soma Diameter	21 μ m
C_m	1 μ F/cm ²	R_a	35.4 Ω -cm
No. of segments ($nseg$)	1	Temperature	35°C
E_{Na}	50 mV	E_{K}	-77 mV
E_{h}	-45 mV		

238 **Table 2: Estimated model parameters**

Parameter	Value	Literature Model Value
$\overline{g_{Kcnj3}}$	$3.5e^{-3}$ S/cm ²	$1e^{-3}$ S/cm ²
$\overline{g_{Kcna1+ab1}}$	0.018 S/cm ²	0.011 S/cm ²
$\overline{g_{Kcnc1}}$	0.018 S/cm ²	0.011 S/cm ²
ϕ_{Kcnc1}	0.2	0.85
$\overline{g_{Cacna1a}}$	0.00005 S/cm ²	$1e^{-5}$ S/cm ²
$\overline{g_{Cacna1b}}$	0.0001 S/cm ²	$1e^{-5}$ S/cm ²
$\overline{g_{Cacna1c}}$	0.006 S/cm ²	$1e^{-5}$ S/cm ²
$\overline{g_{Cacna1d}}$	0.00045 S/cm ²	$1.7e^{-6}$ S/cm ²
$\overline{g_{Cacna1g}}$	0.0003 S/cm ²	$1e^{-5}$ S/cm ²
$\overline{g_{Cacna1i}}$	0.0006 S/cm ²	$2e^{-4}$ S/cm ²
$\overline{g_{HCN1}}$	0.003 S/cm ²	$1e^{-5}$ S/cm ²
$\overline{g_{HCN2}}$	0.009 S/cm ²	$1e^{-5}$ S/cm ²
$\overline{g_{HCN3}}$	0.01 S/cm ²	$1e^{-5}$ S/cm ²
$\overline{g_{HCN4}}$	0.0035 S/cm ²	$1e^{-5}$ S/cm ²
$\overline{g_{Scn1a}}$	0.075 S/cm²	$1e^{-5}$ S/cm ²

Classically, physiological arrest mechanisms exist to establish reliable operation of
the cells' electrical machinery. This ensures the operating points to always be stable.
This naturally embedded property needed to be explicitly modeled in our framework
and was done via additional tuning of channel conductances. In this paper, we report
a single parameter set that was most similar to the experimentally observed
electrophysiological behavior (Table 1 and Table 2).

**Sensitivity Analysis**

Different ion channels underlie different phases of the firing activity such as its
duration, inter-spike interval, extent of depolarization and repolarization. The impact
of these channels on the above features was assessed by running a sensitivity
analysis on the following metrics: firing frequency, AP peak, negative peak of
hyperpolarization, and full width at half maximum (FWHM). For a fixed current
stimulus of 0.1 nA, the conductance of each channel was incrementally varied with
respect to its nominal value (Table 2). For the range of conductances explored, best-
fit linear regression curves were fitted wherein the slope best approximated the
overall trend of the data, thereby capturing the dependency of the metrics on each
ion channel model.

**Modelling and simulation tools**

The model was implemented using NEURON v8.0 (<http://neuron.yale.edu/>) and the
NetPyNE v1.0.0.2 Sobol branch (<http://netpyne.org/>) (Dura-Bernal et al., 2019).
These modelling tools facilitated parallel simulations on high-performance computing
platforms, allowing us to run over 400,000 simulations during model development.

**Data and code availability**

The model source and analysis code were developed as part of the Stimulating
Peripheral Activity to Relieve Conditions (SPARC) program is available on GitHub
([https://github.com/Daniel-Baugh-](https://github.com/Daniel-Baugh-Institute/BiophysicalModellingOfIntrinsicCardiacNeurons)
[Institute/BiophysicalModellingOfIntrinsicCardiacNeurons](https://github.com/Daniel-Baugh-Institute/BiophysicalModellingOfIntrinsicCardiacNeurons)) and the SPARC portal
(<https://doi.org/10.26275/cy9w-ttjn>) under a CC-BY 4.0 license. Model source code is
also available through ModelDB (<https://modeldb.science/2014824>). The model can

also be accessed through the simulation platform oSPARC (<https://osparc.io/>), which
enables users to run simulations with the model using a Graphical User Interface.
The Ten Simple Rules for the Credible Practice of Modelling and Simulation in
Healthcare (Erdemir et al., 2020; Mulugeta et al., 2018) were documented to ensure
data availability and can be found in the Supplementary Material.

**Results**

The model development workflow builds on ion channel expression data along with
public resources on ion channel kinetics towards developing integrated
electrophysiology models of ICNS neurons (Figure 1). HT-qPCR data from single
neurons on ion channel genes were used to select ion channel presence or absence
in single-neuron models (Step I) (Moss et al., 2021). The data were binarized to
represent ion channel presence or absence using a cycle threshold (C_t) cutoff
(Figure 2). Several values for the C_t threshold were considered ranging from 13–17,
but a threshold of 15 cycles was selected so that each neuron included voltage-
gated sodium channels and at least one voltage-gated potassium channel to ensure
the potential for electrical excitability (Edwards et al., 1995; McAllen et al., 2011).
Redundant neurons with the same ion channel combinations were removed to
identify unique neuronal phenotypes (Step II). Corresponding ion channel models
were identified from public databases (Step III). Fixed conductance values were
selected for each (Step IV). Known morphological properties of RAGP neurons were
incorporated to construct a library of parallel conductance models (Step V). Finally,
the model responses to current clamp stimulus were simulated and the firing
properties of each neuronal phenotype were analyzed and classified (Step VI).

**Figure 1: Workflow for development of electrophysiological models starting with**
 **single neuron gene expression data and model database of ion channel**
 **kinetics.**

**Figure 2: Selection of expression threshold for filtering transcriptomics data.**

Number of neurons identified with each ion channel transcript at C_t values from 13–17.

$C_t \leq 15$ was chosen to denote ion channel presence.

Based on thresholding for the apparent presence or absence of the 15 ion channel

genes, coding for 14 different ion channels, from our thresholded transcriptomic

data, we identified 104 unique neuronal phenotypes from 321 sampled neurons

(Figure 3). A maximum of 13 of these ion channels were present in a neuronal

phenotype. Figure 3A shows these ordered from the most commonly (bottom) to

least commonly expressed ion channel gene. Three of the Ca^{2+} channel types

(*Cacna1g*, *Cacna1a*, *Cacna1d*) and two of the HCN types (1,3) were rarely present.

The frequency of each neuronal phenotype was non-uniform (Figure 3B). Fifty-

seven neuronal phenotypes occurred only once; the most common neuronal

phenotype occurred 22 times. 43% (137/321) of neurons belonged to 8 common

types with 14 or more cells in each neuronal phenotype: T4, T7, T14, T15, T23, T30,

T73, T91. While only 79/312 neurons expressed *HCN1*, the commonly occurring

types typically had the *HCN1* channel. Trends in *HCN2*, *HCN3*, and *HCN4* were not

associated with large increases in the number of occurrences of a neuronal

phenotype. We also examined sex-based differences in types and whether they

projected to the sinoatrial node (SAN). Three of the eight common types were from

the female, and T30 was found only in female, non-SAN-projecting neurons. There

were no statistically significant sex-dependent differences in ion channel expression

between the SAN-projecting and non-SAN-projecting RAGP neurons (Moss et al.,

2021).

**Figure 3: Neuronal phenotypes resulting from thresholded single neuron gene**
 **expression data on ion channels. Top:** Binary map for the 104 unique channel
 combinations ordered by frequency of occurrence of ion channels. **Bottom:** Number
 of cells of each neuronal phenotype (321 cells, 104 types). The eight common
 neuronal phenotypes are highlighted in magenta, while the remaining neuronal
 phenotypes are in green.

Responses to current clamp for all neuronal phenotypes matched experimentally
 observed patterns found in multiple species (rodents, minipigs, dogs): phasic
 responses and tonic firing (Figure 4) (J. Andrew Armour, 1991; Edwards et al., 1995;
 Hanna et al., 2021; Harper & Adams, 2021; McAllen et al., 2011). As in experiments,
 electrophysiological responses were dependent on current clamp stimulus strength,
 such that some neurons would transition with increased input current (Figure 4A):
 61% were phasic only (Figure 4B), 24% tonic only (Figure 4C), 11% phasic-to-tonic
 (Figure 4D), 3% were tonic-to-phasic (Figure 4E). The remaining firing patterns
 included stray occurrences of the artificial firing patterns (spontaneous firing,
 incomplete repolarization), which were filtered out and removed from our analysis.
 The primarily phasic and tonic firing patterns emerge from diverse combinations of
 ion channels that contribute to different dynamics in neuronal behaviors.

We used currentscapes (Alonso & Marder, 2019) to visualize the relative contribution
of each ion channel to RMP, inward currents, and outward currents over time in a
representative phasic (Figure S5 A) and tonic (Figure S5 B) neuron. We observed
*Scn1a*, *Cacna1g* and *Cacna1i* to contribute substantially to the active phases of the
action potential, while *Cacna1c* modulated the trough of repolarization. While *HCN3*,
*4* and *Kcnj3* dominated the RMP, *Kcnc1* and inactivating *Kcna1* influenced the
repolarization phase of the spike. In the phasically firing neuron, *Kcnj3* has a much
larger effect on outward currents than in the tonically firing neuron, which had
outward currents dominated by *Kcnc1*.

**Figure 4: Neuronal firing behavior with increasing stimulus strength. (A)** Number
 of neurons with each firing behavior. Example traces for **(B)** phasic (T1), **(C)** tonic

[revised manuscript text omitted]

**Keywords**

autonomic regulation, systems biology, ion channels, Hodgkin-Huxley model, vagus
nerve, intrinsic cardiac nervous system

**Supplemental Data**729 **Supplemental Figures (S1 – S7)**

**Figure S1: Distribution of firing behaviors for C_t thresholds varying from 13 –**746 **17.** The firing behavior of each neuronal phenotype was classified at step input747 stimulus intensities of 0.1, 0.3, and 0.5 nA. Despite the changing C_t threshold values,748 the dominant firing behavior remained phasic. For C_t threshold values ranging from

13 – 16, similar distributions for phasic, tonic, phasic-to-tonic, and tonic-to-phasic

were observed. For $C_t \leq 17$, phasic-to-tonic behavior became more common than

tonic behavior.

Figure S1: Variability in the transcriptomics map for different C_t values explored resulting in the emergence of newer neuronal phenotypes. Neuronal phenotypes based on C_t thresholds varying from 13 – 17. New neuronal types not defined by thresholding transcriptomic data with $C_t \leq 15$ are highlighted in red. Neuronal types are sorted using the same method as in Figure 3, based on the number of genes expressed and the frequency that a gene is expressed. The percent of new neuronal phenotypes for C_t values 13, 14, 16, and 17 is 53%, 47%, 56%, and 77%. While this is a high percentage of the neuronal phenotypes, further

analysis of the frequency of these new neuronal phenotypes revealed that they
 account for only 11-20% of the cells

**Figure S2: Distribution of neuronal phenotype occurrences that were defined**
 **with $C_t \leq 15$ with C_t threshold value varying from 13 – 17.** Neuronal phenotypes
 with more than ten occurrences are highlighted in pink. When using a different C_t
 threshold value, neuronal-types that did not fit into any neuronal-type found with $C_t \leq$
 15 are not shown in the plot. New neuronal transcriptional phenotypes appear at
 different C_t thresholds, but they only account for 11-20% of the cells. We also found
 that all C_t thresholds still produced just 5-8 common neuronal phenotypes with 10 or
 more occurrences containing 25-40% of the cells, suggesting that the frequency of
 occurrences of the neuronal phenotypes remains relatively similar across C_t
 threshold values.

**Figure S3: Comparison of Nav1.1 activation and inactivation curves for**
 **alternative ion channel models.** The models compared are Channelpedia ID #35
 and ModelDB Accession #264834. We identified five alternative Nav1.1 models, of
 which, 3 were unsuitable due to a large window current which required an
 unphysiologically large reversal potential to compensate for the current at rest, or a
 very leaky cell (large R_{in}). To choose between the remaining two models
 (Channelpedia ID #35 and ModelDB Accession #264834), we assessed the
 rheobase. Comparison of the two Na^+ channel candidates in the full parallel
 conductance model identified Channelpedia ID #35 as a better fit.

**Figure S4: Variation of the firing frequency with respect to the ϕ_{Kcnc1} for**
 **different current clamp stimuli surveyed. An optimized best-fit (black line) indicate**
 **an increase of 2.62 Hz per 0.1 increase in fractional amplitude.**

**Figure S5: Currentscapes that depict dynamic contribution of ionic currents**
 **that underly a representative phasic (T1) and tonic (T57) firing profiles at 0.2**
 **nA current clamp stimulus.** Membrane potential is shown in the topmost panel,
 with the contribution of inward ionic currents and outward ionic currents in the next
 two panels, plotted on a semi-logarithmic scale. Overall percent contribution of
 inward and outward ionic currents is depicted as a pie chart shown alongside the
 currentscapes. The currentscapes revealed that *Scn1a* (Nav 1.1) contributes the
 most to inward currents in both neuronal-types, where it contributes to 53.8% of
 inward currents in tonically firing T57 and 30.7% of inward currents in phasically

firing T1. In both T1 and T57, *Cacna1g* (Cav 3.1) and *Cacna1d* (Cav 1.3) sustained
 the depolarized membrane potential. The difference in neuronal behavior between
 the phasically firing T1 and tonically firing T57 is seen largely in the outward currents.
 *Kcnj3* (Kir 3.1) contributes to 74.8% of outward currents in phasically firing T1,
 whereas outward currents in tonically firing types are primarily due to *Kcnc1* (Kv 3.1).

**Figure S6: Effect of varying the Na^+ channel inactivation parameter (h_∞) on**
 **electrophysiological behavior.** Simulations are shown for a +20% change in h_∞ (A
 and B) and a -20% change in h_∞ (C and D). A C_t threshold of 15 was used so that
 results for the electrophysiological behavior of neuronal phenotype T1 can be
 compared to the results in Figure 4, which shows that T1 has phasic behavior at all
 three stimulus intensities. h_∞ values ranged within the range of what would be
 expected for inter-model variability.

Reviewer 2

The study is motivated by a fascinating idea, which is, being able to explicitly include information about patterns of ion channel expression into an excitable membrane model, in this case, from intrinsic cardiac neuron transcriptomics. However, the models were chosen from databases that assume channel dynamics, in particular gating dynamics, that by their formulation bias the predictions in at least a couple of ways. The most important one is the use of powers in the gating terms, which impact the sizes of the maximal conductances of the model. One key question that arises from this is: are the ratios of Na and K maximal conductances equivalent to the ratios of the levels of expression from the data corresponding to Na and K channels? The models chosen for the study, as presented, predict ratios of 3-5 times as many Nav1.1 channels as Kv1.1 channels, for instance. Is this accurate? I would very much like to see some evidence to that effect. Electrophysiological recordings and other splice variant data indicate the opposite (several fold more K channels in comparison to Na channels, for instance), but that is not conclusive data that applies to all species and neurons within them.

Regarding originality, the question has been asked before (e.g. Herrera-Valdez, et al. 2013, Nowotny et al. 2007), but to my knowledge, there have not been any studies in which transcriptomic or proteomic data have been used. In this regard, the study is motivated by a very interesting paradigm linking modelling and experimental measurements, but again, choose arbitrarily from model databases arguing that many people use those models (see comments about powers in gating terms in annotated pdfs and above). One important fact to consider is that the behaviour of the neurons of interest in the study can be captured initially with a simple 2D model with voltage-gated Na and K-delayed rectifier channels. The conclusions from using a model with high dimensionality are hard to interpret and it is easy to conclude, wrongly, that "wild combinations of parameters indicate wild combinations of expression levels", as suggested by some. Highly dimensional models may show combinations of parameters that may not be correct in light of physiological considerations. Also, the authors choose to model ionic currents with different functional forms (fundamentally different assumptions for the underlying biophysics) within the same model, which does not help for interpretation.

As to the dynamics of the model, since there seems to be a gradient of behaviours that stem all from phenomena captured by 2D biophysics, I think the authors could have focused on building up from simple models to tackle the issues of diversity in the neuronal population, to network dynamics as mentioned in the discussion of the paper. The model does not go significantly further in making predictions, in comparison to the transcriptomics data, but analysing the dynamics from a tractable perspective would have been more informative, and then the inclusion of more ion channels to explain how redundancies occur would have been appropriate.

In spite of the poor modelling and choices, and having done no geometrical or other kinds of analysis that are more meaningful than the statistics, I believe that the approach heads fundamentally in the right direction, and other modelling approaches could complement what is reported in this study, provided that the transcriptomic data is shared properly (not just the binary data about whether a channel gene is transcribed or not).

Response:

We appreciate the detailed insight and critical feedback to improve the manuscript and address the critiques as below. We agree that the strength of this work lies in exploring the premise that transcriptomic data on ion channel expression can explicitly be included in an excitable membrane model. We have addressed the reviewer's major concerns stated in the above paragraphs in the section below and address each piece of specific feedback after.

We agree with the reviewer's concern regarding how gating parameter powers can influence ion channel dynamics and electrophysiological firing patterns.

Action Taken:

We have added an explanation of our rationale for choosing Hodgkin-Huxley models in the Discussion section. The text starts with: "The models were selected from three public databases: Channelpedia (Ranjan et al., 2011), ModelDB (McDougal et al., 2017), and Ion Channel Genealogy (Podlaski et al., 2017)... may be used without any loss of general function."

Response:

We have addressed the reviewer's concern about the ratio of Na and K channels by including an analysis showing that the relative expression of Na:K is 3.4 on average compared to the 4.2-fold difference in conductance values predicted by the model.

Action Taken:

We have included a short discussion of the limited protein expression data available, which indicates that Na channel expression is higher than K channel expression. This can be found in the Discussion section.

Response:

The shortlisted ion channels from the databases were further screened by comparing their activation and inactivation kinetics (mh curves).

Action Taken:

We have added the following sentence in the Methods section: "Multiple ion channel models of the same genotype, mined from the different databases, were compared on the basis of their activation and inactivation dynamics to assess their regions of operation."
In addition to selecting Na and K models that were already part of the Methods section, we have added a paragraph on how voltage-gated Ca channels were shortlisted. We have expanded Figure 4 to demonstrate the mh curves for all the voltage-gated sodium and calcium channels explored.

Response:

The reviewer makes a good point about the high dimensionality of the model making it more difficult to interpret the results. Our study addresses this directly by showing that variability in the combinations of ion channels converges at the physiological scale to yield lower dimensionality of phenotypes (as defined by a particular response to stimulus). Regarding the concern about high dimensional models containing parameters that may be unphysiological, the relative conductances used in the model were correlated with the relative expression levels in the

transcriptomic data. An analysis of the correlation between relative conductance and relative expression was performed as result of reviewer feedback (thanks!). We would like to note that transcriptomic data were not used to constrain the parameter values. The analysis showed that model parameter values correspond with physiological measurements.

Action Taken:

This analysis was added as Figure 8 in the revised manuscript.

Response: *We agree with the reviewer that we have employed different functional forms of the calcium channel model. Our understanding is that each variant of calcium channels has a distinct and unique gating property. These channels differ with respect to their threshold of activation and speed of activation. Additionally, not all variants inactivate, and if they do, do not inactivate to the same extent and by the same gating variable. Owing to this variability, a single functional form could not be employed for all calcium channels.*

Action Taken: *In the manuscript, we have explicitly mentioned how we attempted to ensure biophysical viability of the calcium models chosen in the Methods and Discussion.*

Response:

The reviewer makes an excellent point that an alternative to our approach of simulating all of the ion channels found in the transcriptomic data would have been to start from a two ion channel model and progressively add channels to explain how channel redundancies occur. We believe that both approaches are rigorous and informative. We opted for simulating all of the ion channels together because the neuronal genotype with the fewest ion channels has one sodium and one potassium channel. About 38% of our neuronal genotypes exhibited a firing pattern that was invariant to parametric variations. Due to this emergent difference in response to parametric variations, exploring a mosaic of ion channel combinations across the 104 neuronal genotypes was more appropriate to survey the range of physiological firing patterns, as depicted in Figure 5. Restricting our neuronal-genotype to 2 ion channel models (T104) would result in a consistent phasic firing pattern which would be representative of 38% of the entire neuronal cohort. By comparing the firing behaviour of different neuronal genotypes, we are able to examine ion channel redundancies.

Action Taken: *We have added a sentence at the end of the introduction acknowledging this alternative way to frame the manuscript as an equally valid alternative.*

Response:

Thank you for pointing out that we did not include a link to the original transcriptomic dataset used for model development. The transcriptomic data are available on the SPARC portal (<https://doi.org/10.26275/z6jn-j5tx>).

Action Taken: *A sentence directing the reader to this resource has been added to the Data and code availability subsection in the Methods.*

Specific comments

Highlighted manuscript text: The parameters of the ion channel models were grounded based on passive properties: resting membrane potential, input impedance, and rheobase

Comment: What do passive properties have to do with the active properties conferred by the channels?

Response: *The reviewer is correct in pointing out that the passive properties are not directly related to the active properties conferred by the channels. We constrained the model based on these passive properties because we did not want to bias our result or overfit the model in any way.*

Action Taken: *We now clarify this in the Abstract. “The parameters of the ion channel models were grounded based on passive properties (resting membrane potential, input impedance, and rheobase) to avoid biasing the dynamic behaviour of the model.”*

Highlighted manuscript text: The single-cell transcriptomic data formed the basis for ion channel combinations in each neuronal model

Comment: How? What is the direct relationship between the transcriptomic data and the ion channel combinations in the model?

Response and Action Taken: *We have clarified this bullet point to add more information on the relationship between the transcriptomic data and the ion channel combinations. The bullet point now reads: “The single-cell transcriptomic data were thresholded to select the ion channel combinations in each neuronal model”.*

Highlighted manuscript text: Heterogeneity in the electrophysiological behavior of neurons is driven by variation in ion channel expression

Comment: This is obvious without the model. What are the patterns or mechanisms underlying the heterogeneity as explained by the model?

Response: *The reviewer is correct that this statement is obvious without the model.*

Action Taken: *We have replaced the bullet point with a new one highlighting a result that was spurred by reviewer feedback. The new bullet point is: “The ratios of model-predicted conductance values are correlated with the gene expression ratios from transcriptomic data.”*

Highlighted manuscript text: we use the well-established Hodgkin-Huxley models and combine them with single-cell transcriptomics data to identify specific ion channel combinations for each neuron

Comment: These models are well known for not being able to reproduce firing patterns in neurons with accuracy in regard to ion channel expression or ion current data. Part of the problem is the powers in the gating, which cause the values of the maximal conductances to misrepresent the sizes of the ion channel populations. Also, the model used in this paper has a multitude of functional forms for different ion channels, all voltage-gated or ligand-gated, which introduces an unwanted source of variability.

Response: *This is a fair point that there are certainly limitations that come with using Hodgkin-Huxley models. We found that in our case we were able to reproduce reasonable firing patterns with correlations between ion channel expression and model conductances. The maximal ion channel conductances were selected based on what values produced reasonable resting membrane potential, input impedance, rheobase, and leak reversal potential. We found that the model-predicted conductances were correlated to the ion channel expression level, which was not used to select model conductances. Furthermore, other ion channel conductances predicted by models tuned to electrophysiological data have been found to be correlated with ion channel expression levels (Nandi et al., 2022).*

Action Taken: *An additional figure (Fig. 8) has been added to address this point (see responses to similar comments below for more details and figure). We have also additionally explained the pecking order of the public databases and our approach at initial screening of the models available.*

Highlighted manuscript text: neurons of the same cell type have electrophysiological behavior consistent with each other in response to current clamp stimulus, but vary in their ion channel conductance densities

Comment: Please provide evidence for this statement in the context of this paper.

Response: *This sentence paraphrases a point made in Goillard and Marder 2021, about how neurons of the same cell type have variability in ion channel conductance densities while showing similar behavior. In the context of our work, ICN neurons have been shown to be electrically excitable with phasic or tonic responses (McAllen et al., 2011). Our transcriptomic data combined with the models indicate that there are many combinations of ion channels that lead to phasic behaviour and many combinations that lead to tonic behaviour.*

Action Taken: *We have added a reference to this sentence.*

Highlighted manuscript text: Goillard & Marder, 2021

Comment: Yes, but see Nowotny, 2007. Tails wagging the dogs ([PDF] mit.edu)

Response: We stated that ion channel heterogeneity may contribute to the observation that neurons of cell type can have consistent electrophysiological behavior in response to one stimulus but a variety of responses in response to a different input (Goaillard & Marder, 2021). If we understand correctly, the reviewer is saying that rather than ion channel heterogeneity, the phasic versus tonic behaviour are a result of bifurcations in the stability of the dynamical system (Nowotny & Rabinovich, 2007).

We agree with the reviewer's viewpoint. There have been instances where a parameter shift in single-neuronal Hodgkin-Huxley model has resulted in varied firing characteristics (Guckenheimer & Labouriau, 1993; Rush & Rinzel, 1995; Izhikevich, 2003; Doi & Kumagai, 2005; Postnova et al., 2007)). We have observed 64/104 of our neuronal genotypes models to exhibit both tonic and phasic firing based on stimulus strength and ion channel conductances.

However, the firing characteristics of single-neurons are rapidly adjusted when in a network. In their model, Nowotny states that the "transformations of the phase portrait depend only on one control parameter, i.e., the equal strength of the couplings". In vivo, the principal neurons form a network within the RAGP wherein the 64/104 phasic and tonic neurons will interconnect with the 40/104 robustly phasic neurons, and non-excitable small intensely fluorescent cells (Hanna et al., 2021; Moss et al., 2021). The individual firing characteristics of these single neurons will thereby be mediated by the strength of their connections as well as the inputs received from vagus nerve. While the strength of the couplings does not apply for single-neuron models (Nowotny & Rabinovich, 2007), a bifurcation analysis framework may be more effective as we extend this work to examine our principal neuron models in a network.

Action Taken: We have added information on bifurcation analysis of the planned network model as part of the Discussion, starting with "Several planned extensions of the current model...."

Highlighted manuscript text: We present a strategy for using single-neuron transcriptomic data to predict neuronal membrane physiology, demonstrating a workflow for building a library of neuronal phenotype models

Comment: This is extremely interesting. However, do you really match the relative ratios of pair-wise ion channel expression with the dynamics in the model and the corresponding pair-wise ratios of maximal conductances corresponding to those ion channels?

Response: This is a very interesting point that we had not fully considered in the manner you suggest, and we thank you for bringing it up.

Action Taken: We calculated the median expression fold differences compared to *Cacna1a*, the ion channel with the lowest conductance in the model. The median fold difference was calculated after normalizing all gene expression using the housekeeping gene *GAPDH* (ΔCt).

The fold difference for each gene compared to *Cacna1a* was calculated ($\Delta\Delta Ct = \Delta Ct_{gene} - \Delta Ct_{Cacna1a}$), then used to calculate fold difference in expression. The model-predicted ratio of conductances were correlated with the ratio of expression levels in the transcriptomic data ($R^2=0.67$). There are some exceptions to the correlation, notably *Hcn3* and *Cacna1b* which were removed for the correlation analysis. It should be noted that the maximal conductance values were selected based on the passive electrical properties of the neurons, not the transcriptomic data. Taken together, these findings suggest that the model predicted conductances independently correlate with relative expression of ion channels. The figure below (Fig. 8 in the manuscript) and additional text discussing the correlation between the expression and conductance values have been added to the results section.

Highlighted manuscript text: used the Goldman-Hodgkin-Katz

Comment: Why is this a good idea for Ca channels and not for Na or K voltage-gated channels? They all gate with voltage and they all conduct ions...

The nonlinear contributions from the driving force terms in the GHK model can differ greatly from other ion-channel contributions, shifting the balance between channels predicted by the model.

Response: We agree with the reviewer's comment that the GHK equation applies to Ca, Na and K voltage-gated channels. Intra and extracellular potassium and sodium concentrations are tightly regulated during a spike such that they are maintained within the same order of magnitude, which results in their Nernst Potentials to remain constant. On the other hand, intracellular calcium concentration rises ~10-fold during a spike. Extracellular calcium

concentration is of the order of mM, while basal intracellular calcium concentration is of the order of 100s nM. Calcium transient, which underlies an action potential, causes intracellular calcium concentration to rise to the order of μM , which results in a ~ 30 mV change in its Nernst Potential. To ensure that this change in electrochemical driving force is accounted for in our models, we explicitly incorporated the GHK model for Ca channels.

Action Taken: We have added this explanation in the Methods section.

Highlighted manuscript text: We refer to each unique combination of ion channels as a neuronal phenotype.

Comment: Not necessarily accurate in the sense that similar electrophysiological or otherwise behaviours could be displayed by neurons having the same “electrophysiological phenotype”

Response: You are correct, neuronal genotype would be more accurate for what we are describing.

Action Taken: We have changed this nomenclature throughout the manuscript.

Highlighted manuscript text: *Kcna1* (Kv 1.1) is a delayed rectifier potassium (KDR) channel, which is non- or slowly-inactivating (timescale of seconds) (Song, 2002). Our data set showed robust expression of *Kcna1* (.. subunit of Kv 1.1). There was also a dominant expression of *Kcnab1* ($\beta 1$ regulatory subunit) across all neuronal phenotypes

Comment: What is the range of ratios of Nav1.1 to Kv1.1 channels in the neurons? Is it larger than 1? What are the corresponding ratios of maximal conductances in the corresponding neurons? if those two quantities do not match, the approach is flawed at some point. My guess would be the choice of models.

Response:

Using the same methodology as previously described to calculate relative expression levels, we found that the *Scn1a* was expressed on average at 3.4 (median 2.7) times greater levels than *Kcna1* in each of the neurons. This ratio is in agreement with the maximal conductance ratio for Nav1.1 and Kv1.1, which was 4.2.

On the protein expression side, there is limited data that suggests Nav1.1 expression is about 1.4 times greater than Kv1.1 expression (Gu et al., 2018). These findings are from mammalian central neurons, as measurements in mammalian peripheral neurons could not be found. Furthermore, single channel conductance for Nav1.1 is roughly 1.5 times higher than Kv1.1 (17 pS for Nav1.1 compared to 12 pS for Kv1.1) (Vanoye et al., 2006; Streit et al., 2014). Thus, the combination of 1.4 fold expression and 1.5 fold channel conductance results in an

approximately 2 fold higher conductance for Nav1.1. These data also suggest that our conductance ratio for Nav1.1 and Kv1.1 is reasonable.

Action Taken:

Information on the relative conductance ratios is included in the results section in the text corresponding to Figure 8. The literature on protein expression of Nav1.1 versus Kv1.1 has been added to the Discussion section.

Highlighted manuscript text: *degenerate parameter set*

Comment: what made those parameter sets degenerate?

Response: *The reviewer is correct in pointing out that just because these parameter sets had physiologically reasonable input impedance, reversal potential, and rheobase doesn't mean they are degenerate.*

Action Taken: *The label degenerate has been removed.*

Highlighted manuscript text: *Figure 5 comment (currently Fig. 7 in the manuscript)*

Comment: Show an enhancement of the curve for the range before 0.1 nA in the same or in a different graph

Response and Action Taken: *We have added this inset to the figure.*

Comment: The models chosen for this study have one big flaw. The prediction of the number of Nav channels is going to be affected heavily by the powers in the gating terms. This is an inherited feature from the HH model but it can be corrected. Is it really the case that the expression of Nav1.1 channels in a single neuron is 3-4 times the expression of Kv1.1 channels, and not the opposite?

Response: *As mentioned in the previous response, the qPCR data and protein data support this model prediction. It has been reported that Nav1.1 expression is higher than Kv1.1 expression (Gu et al., 2018). The qPCR data show that the mean Scn1a expression was 3.4 times higher than Kcna1 expression on a cell by cell basis. The minimum expression ratio was 1.4 and the maximum was 50.4. This expression ratio and corresponding maximal conductance ratio was not surprising to us given that similar ratios of maximal conductances for sodium and*

KDR channels have been reported previously (Rybak et al., 1997; Rogers et al., 2000; Yaghini Bonabi et al., 2014).

Action Taken: *Additional text addressing these concerns has been added to the Discussion section in the “Strengths and limitations of the model” subsection.*

Comment: The discussion in lines 446-464 approximately is highly relevant to the observations made above in regard to the sizes of the maximal conductances. However, even if the assumptions for the gating parameters were justified, the powers and their effect on the gating variables yield radically different predictions about the Na/K complements of channels.

Response: *As stated in the lines of the Discussion section highlighted, we agree that there are limitations to translating transcriptomic data to electrophysiological dynamics.*

Action Taken: *We have expanded upon the Discussion section to include text on the available protein data that suggest that Nav1.1 expression may be higher than Kv1.1 expression. We also added text on how the translation of ion channel expression levels to channel dynamics is further complicated by the different single channel conductances of different ion channels.*

References

- Doi S & Kumagai S (2005). Generation of very slow neuronal rhythms and chaos near the Hopf bifurcation in single neuron models. *J Comput Neurosci* **19**, 325–356.
- Goaillard J-M & Marder E (2021). Ion Channel Degeneracy, Variability, and Covariation in Neuron and Circuit Resilience. *Annu Rev Neurosci* **44**, 335–357.
- Guckenheimer J & Labouriau JS (1993). Bifurcation of the Hodgkin and Huxley equations: A new twist. *Bull Math Biol* **55**, 937–952.
- Gu Y, Servello D, Han Z, Lalchandani RR, Ding JB, Huang K & Gu C (2018). Balanced activity between Kv3 and Nav channels determines fast-spiking in mammalian central neurons. *iScience* **9**, 120–137.
- Hanna P et al. (2021). Innervation and Neuronal Control of the Mammalian Sinoatrial Node a Comprehensive Atlas. *Circ Res* **128**, 1279–1296.
- Izhikevich EM (2003). Simple model of spiking neurons. *IEEE Trans Neural Netw* **14**, 1569–1572.
- McAllen RM, Salo LM, Paton JFR & Pickering AE (2011). Processing of central and reflex vagal drives by rat cardiac ganglion neurones: an intracellular analysis. *J Physiol* **589**, 5801–5818.
- Moss A, Robbins S, Achanta S, Kuttippurathu L, Turick S, Nieves S, Hanna P, Smith EH, Hoover DB, Chen J, Cheng ZJ, Ardell JL, Shivkumar K, Schwaber JS & Vadigepalli R (2021). A single cell transcriptomics map of paracrine networks in the intrinsic cardiac nervous system. *iScience* **24**, 102713.
- Nandi A, Chartrand T, Van Geit W, Buchin A, Yao Z, Lee SY, Wei Y, Kalmbach B, Lee B, Lein E, Berg J, Sömböl U, Koch C, Tasic B & Anastassiou CA (2022). Single-neuron models linking electrophysiology, morphology, and transcriptomics across cortical cell types. *Cell Rep* **40**, 111176.
- Nowotny T & Rabinovich MI (2007). Dynamical origin of independent spiking and bursting activity in neural microcircuits. *Phys Rev Lett* **98**, 128106.
- Postnova S, Voigt K & Braun HA (2007). Neural synchronization at tonic-to-bursting transitions. *J Biol Phys* **33**, 129–143.
- Rogers RF, Rybak IA & Schwaber JS (2000). Computational modeling of the baroreflex arc: nucleus tractus solitarius. *Brain Res Bull* **51**, 139–150.
- Rush ME & Rinzel J (1995). The potassium A-current, low firing rates and rebound excitation in Hodgkin-Huxley models. *Bull Math Biol* **57**, 899–929.
- Rybak IA, Paton JFR & Schwaber JS (1997). Modeling Neural Mechanisms for Genesis of Respiratory Rhythm and Pattern. I. Models of Respiratory Neurons. *J Neurophysiol* **77**, 1994–2006.
- Streit AK, Matschke LA, Dolga AM, Rinné S & Decher N (2014). RNA editing in the central cavity as a mechanism to regulate surface expression of the voltage-gated potassium

channel Kv1.1. *J Biol Chem* **289**, 26762–26771.

Vanoye CG, Lossin C, Rhodes TH & George AL Jr (2006). Single-channel properties of human Nav1.1 and mechanism of channel dysfunction in SCN1A-associated epilepsy. *J Gen Physiol* **127**, 1–14.

Yaghini Bonabi S, Asgharian H, Safari S & Nili Ahmadabadi M (2014). FPGA implementation of a biological neural network based on the Hodgkin-Huxley neuron model. *Front Neurosci* **8**, 379.

Dear Professor Vadigepalli,

Re: JP-RP-2024-287595R1 "Biophysical Modelling of Intrinsic Cardiac Nervous System Neuronal Electrophysiology based on Single-cell Transcriptomics" by Suranjana Gupta, Michelle M Gee, Adam John Hunter Newton, Lakshmi Kuttippurathu, Alison Moss, John D. Tompkins, James S Schwaber, Rajanikanth Vadigepalli, and William Lytton

We are pleased to tell you that your paper has been accepted for publication in The Journal of Physiology.

Yours sincerely,

Natalia Trayanova
Senior Editor
The Journal of Physiology

If you would like to receive our 'Research Roundup', a monthly newsletter highlighting the cutting-edge research published in The Physiological Society's family of journals (The Journal of Physiology, Experimental Physiology, Physiological Reports, The Journal of Nutritional Physiology and The Journal of Precision Medicine: Health and Disease), please click this link, fill in your name and email address and select 'Research Roundup':
<https://www.physoc.org/journals-and-media/membernews>

- You can help your research get the attention it deserves! Check out Wiley's free Promotion Guide for best-practice recommendations for promoting your work at: www.wileyauthors.com/eoo/guide. You can learn more about Wiley Editing Services which offers professional video, design, and writing services to create shareable video abstracts, infographics, conference posters, lay summaries, and research news stories for your research at: www.wileyauthors.com/eoo/promotion.

EDITOR COMMENTS

Reviewing Editor:

Thank you for thoroughly addressing the critiques.